# TRPM7 is an essential regulator for volume-sensitive outwardly rectifying anion channel

Tomohiro Numata [1], Kaori Sato-Numata[1,2], Meredith C. Hermosura[3], Yasuo Mori[4] &
Yasunobu Okada [5,6,7 ✉]

Animal cells can regulate their volume after swelling by the regulatory volume decrease (RVD) mechanism. In epithelial cells, RVD is attained through KCl release mediated via volume-sensitive outwardly rectifying $Cl^-$ channels (VSOR) and $Ca^{2+}$-activated $K^+$ channels. Swelling-induced activation of TRPM7 cation channels leads to $Ca^{2+}$ influx, thereby stimulating the $K^+$ channels. Here, we examined whether TRPM7 plays any role in VSOR activation. When TRPM7 was knocked down in human HeLa cells or knocked out in chicken DT40 cells, not only TRPM7 activity and RVD efficacy but also VSOR activity were suppressed. Heterologous expression of TRPM7 in TRPM7-deficient DT40 cells rescued both VSOR activity and RVD, accompanied by an increase in the expression of LRRC8A, a core molecule of VSOR. TRPM7 exerts the facilitating action on VSOR activity first by enhancing molecular expression of LRRC8A mRNA through the mediation of steady-state $Ca^{2+}$ influx and second by stabilizing the plasmalemmal expression of LRRC8A protein through the interaction between LRRC8A and the *C*-terminal domain of TRPM7. Therefore, TRPM7 functions as an essential regulator of VSOR activity and LRRC8A expression.

[1] Department of Physiology, Graduate School of Medical Sciences, Fukuoka University, Fukuoka, Japan. [2] Japan Society for the Promotion of Science, Tokyo, Japan. [3] John A. Burns School of Medicine, Honolulu, HI, USA. [4] Laboratory of Molecular Biology, Department of Synthetic Chemistry and Biological Chemistry, Graduate School of Engineering, Kyoto University, Kyoto, Japan. [5] National Institute for Physiological Sciences, Okazaki, Japan. [6] Department of Physiology, School of Medicine, Aichi Medical University, Nagakute, Japan. [7] Department of Physiology, Kyoto Prefectural University of Medicine, Kyoto, Japan. ✉email: okada@nips.ac.jp

Animal cells regulate their cell volume even under hypotonic stress (due to decreased extracellular osmolarity or increased intracellular osmolarity) by displaying regulatory shrinking activity, termed the regulatory volume decrease (RVD), after passive osmotic cell swelling. The RVD response is attained through the release of water driven by channel- and transporter-mediated KCl efflux (see Reviews:[1,2]). In epithelial cells, volume-regulatory KCl release was shown to primarily result from parallel activation of the volume-sensitive $Cl^-$ channels and $Ca^{2+}$-activated $K^+$ channels[3–5], the latter identified later as IK1[6]. Activation of these volume-regulatory channels is preceded by activation of a type of nonselective cation channels[7–9], which was recently identified as TRPM7 in human epithelial cells and shown to be induced by membrane stretch associated with osmotic cell swelling[10]. Stretch-induced activation of $Ca^{2+}$-permeable TRPM7 causatively leads to opening of $Ca^{2+}$-activated $K^+$ channels by increasing the intracellular free $Ca^{2+}$ level[10]. While it is thus clear that there is a functional coupling between TRPM7 and $Ca^{2+}$-activated $K^+$ channels, the possibility of functional coupling between TRPM7 and the volume-sensitive $Cl^-$ channel has not been studied.

The volume-sensitive $Cl^-$ channel was discovered in 1988 independently by two groups[3,11] and is called the volume-sensitive outwardly rectifying anion channel (VSOR)[12], the volume-regulated anion channel[13], or the volume-sensitive organic osmolyte-anion channel[14] with exhibiting common phenotypical properties. Recently, a leucine-rich repeat (LRR)-containing protein, LRRC8A, was identified as an essential core component of VSOR through unbiased genome-wide approaches[15,16]. It was also shown that functional VSOR activity requires that LRRC8A forms multimers with LRRC8C, 8D, or 8E[16–18], creating functional heteromeric channels of various subunit configurations. However, there remains a possibility that an as-yet-undetermined pore-related component or regulatory subcomponent other than LRRC8 members is a prerequisite to VSOR activity (see Reviews:[19,20]). Also, it is highly feasible that LRRC8A/C/D/E interacts with other protein(s), to form a fully functioning VSOR, because the LRR motif is known to mediate protein–protein interactions[21]. Thus, it is intriguing to examine whether TRPM7 exhibits any molecular interaction with LRRC8A, thereby facilitating the RVD process.

In the present study, we used electrophysiological and genetic approaches to investigate functional and molecular interactions between VSOR/LRRC8A and TRPM7 in human epithelial cell-derived HeLa cells and chicken B cell-derived DT40 cells. We demonstrate that TRPM7 expression is essential for LRRC8A expression and VSOR activity and, thereby playing a crucial prerequisite role in the RVD process.

## Results

### TRPM7 regulates VSOR activity and LRRC8A expression in human epithelial HeLa cells.

To examine the functional linkage between TRPM7 and VSOR activity, we first performed patch-clamp measurements of VSOR $Cl^-$ channel activity in mock-transfected and TRPM7-siRNA-treated human epithelial HeLa cells, since HeLa cells endogenously express both $Mg^{2+}$-sensitive TRPM7 channels[10] and DCPIB-sensitive VSOR channels[22]. Under $Cs^+$-rich conditions that eliminate $K^+$ channel currents, hypotonic stimulation of mock-transfected control cells induces activation of anionic currents with biophysical properties characteristic of VSOR $Cl^-$ channel (see Reviews:[12–14]): instantaneous activation elicited by voltage pulses, followed by deactivation time course at high positive membrane potentials (Fig. 1a), and an outwardly rectifying current–voltage (I–V) relationship (Fig. 1b). The VSOR whole-cell currents were markedly suppressed in TRPM7-siRNA-treated HeLa cells (Fig. 1a, b). At +100 and

−100 mV, the currents were suppressed to $50.0 \pm 9.8$ and $52.3 \pm 7.6\%$ ($n = 5$), respectively, broadly consistent with the decrease in TRPM7 mRNA expression, down to $45.3 \pm 5.1\%$ ($n = 5$), shown by RT-PCR (Fig. 1c and Supplementary Fig. 3a) and observed in previous studies[10,23,24]. These results suggest that the endogenous VSOR activity in HeLa cells is functionally regulated by TRPM7 expression.

Because LRRC8A has been described as an essential component of VSOR channels in human cells[15,16], mouse cells[25], and rat cells[26,27], we sought to determine if there is a relationship between TRPM7 and LRRC8A. Using RT-PCR, we explored the possibility of the molecular linkage between TRPM7 and LRRC8A expression. Indeed, we found that expression of hLRRC8A mRNA was reduced to $59.4 \pm 3.3\%$ ($n = 5$) in HeLa cells treated with TRPM7-siRNA (Fig. 1c and Supplementary Fig. 3a). The downregulation of LRRC8A expression in TRPM7-silenced HeLa cells supports the existence of a molecular relationship between TRPM7 and LRRC8A that, at least in part, be responsible for the observed functional coupling between TRPM7 and VSOR activity in HeLa cells.

In contrast to TRPM7, it appears that another mechano-sensitive TRPM member, TRPM4[28], is not involved in regulation of VSOR activity and LRRC8A expression, because siRNA-mediated knockdown of endogenous TRPM4, as shown in Supplementary Fig. 1, failed to affect the VSOR whole-cell currents (a, b) and the endogenous expression of hLRRC8A (c) in HeLa cells.

To explore whether the regulatory role of TRPM7 is mediated by steady-state $Ca^{2+}$ influx through the plasmalemmal TRPM7 channel, effects of 2-day treatment with a TRPM7 blocker NS8593[29] or EGTA were observed in HeLa cells. Such long-time exposure to NS8593 (30 μM), as shown in Fig. 2 (NS8593), actually not only reduced TRPM7 currents (a, b) and the cytosolic free $Ca^{2+}$ level (c) to the extent similar to the $Ca^{2+}$ level reduced by TRPM7 knockdown (c: siRNA) but also largely suppressed expression of LRRC8A mRNA (d; Supplementary Fig. 3b) and whole-cell VSOR currents (e) again to the extent similar to the VSOR currents reduced by TRPM7 knockdown (e: siRNA). Treatment with extracellular EGTA (2 mM) for 2 days, as shown in Fig. 2 (EGTA), duplicated the suppressive effect of NS8593 (NS8593) on the cytosolic $Ca^{2+}$ level (c), LRRC8A mRNA expression (d; Supplementary Fig. 3b), and VSOR currents (e). Thus, it appears that TRPM7-induced regulation of VSOR activity and LRRC8A expression is mediated by steady-state $Ca^{2+}$ influx via TRPM7 channels, which is involved in maintenance of the control level of cytosolic free $Ca^{2+}$.

### Functional expression of VSOR and TRPM7 in chicken DT40 cells.

To further investigate the functional and molecular interactions between TRPM7 and VSOR, we next used the chicken B-cell line, DT40, commonly used to study gene disruption and recombination because of their high ratio of targeted random recombination[30]. First, we confirmed the endogenous activities of VSOR and TRPM7 in DT40 cells by patch-clamp experiments. As shown in Fig. 3a, the whole-cell $Cl^-$ current slowly developed after applying a hypotonic solution. The current responses to voltage step pulses showed instantaneous activation followed by slow inactivation kinetics at high positive membrane potentials (Fig. 3a: B) as well as an outward-rectifying I–V relationship (Fig. 3b: Hypo). In addition, the swelling-activated whole-cell current was blocked by a VSOR-specific blocker, DCPIB[31], added to the bath solution (Fig. 3a: C, b: +DCPIB). These electrophysiological and pharmacological properties are characteristic of VSOR channels and identical to those observed in human epithelial HeLa cells. On the other hand, endogenous TRPM7

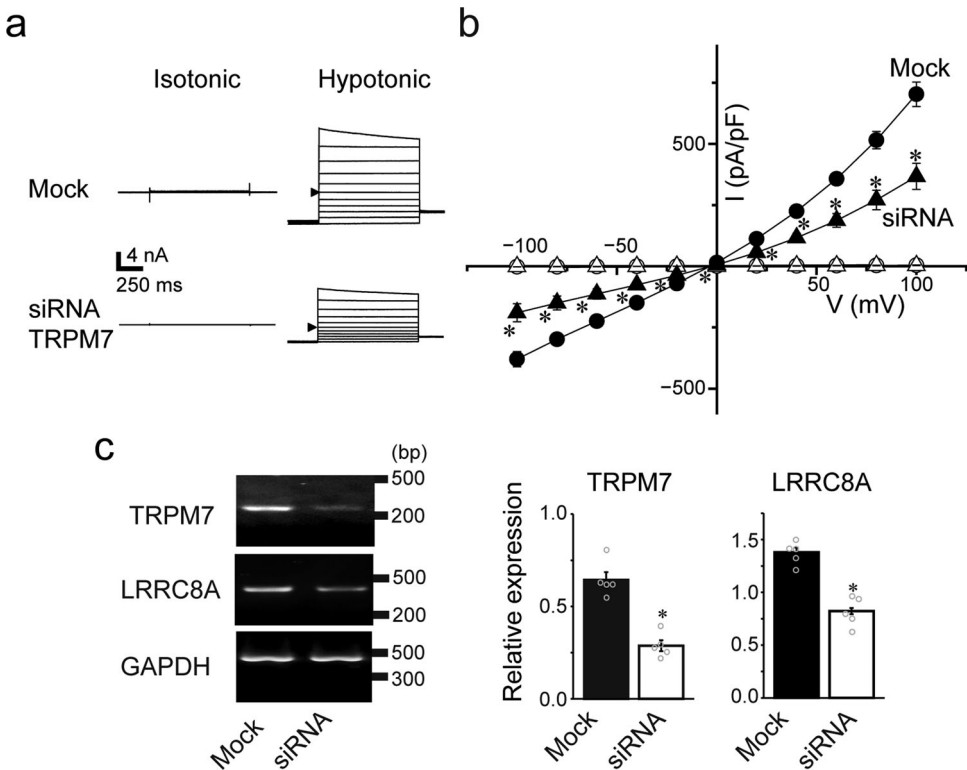

**Fig. 1 TRPM7 expression is involved in VSOR activity and LRRC8A expression in HeLa cells. a** Representative whole-cell current traces elicited by step pulses in isotonic and hypotonic conditions in Mock-transfected (*Mock*) and siRNA-TRPM7-treated (*siRNA*) cells. Arrowheads indicate currents at 0 mV. **b** Mean current (*I*)–voltage (*V*) relationships for swelling-activated whole-cell currents observed in *Mock* and *siRNA* cells ($n = 12$–18). Each data point represents the mean ± SEM (vertical bar) of *n* samples. *$P$ (=0.00053, 0.00036, 0.00028, 0.00024, 0.00033, 0.0018, 0.000083, 0.000085, 0.000091, 0.00010, and 0.00011 at −100, −80, −60, −40, −20, 0, +20, +40, +60, +80, and +100 mV, respectively) <0.005 compared to the *Mock* cell data by *t*-test. **c** Effects of TRPM7 knockdown on expression of TRPM7 and LRRC8A mRNAs. Left panel shows the PCR products from *Mock* or *siRNA* cells for TRPM7, LRRC8A, and the constitutively transcribed control, GAPDH. The nucleotide sequences of the PCR products obtained with TRPM7- and LRRC8A-specific primers were completely identical to the corresponding sequences for TRPM7 (human: NM_017672) and LRRC8A (human: NM_001127244), respectively. Right panels show the bar graph representation of the relative expression values of the optical densities in pixels of the PCR bands of TRPM7 (left bars: 45.3 ± 5.1%) and LRRC8A (right bars: 59.4 ± 3.4%) in *siRNA* cells compared to those in *Mock* cells. The values were calculated from five independent PCR amplifications after normalization to the corresponding band of GAPDH control. *$P$ (=0.00013 and 0.00010 for *TRPM7* and *LRRC8A*, respectively) <0.0005 compared to *Mock* cell data by *t*-test.

activity in DT40 cells was revealed by removal of extracellular divalent cations even under isotonic conditions (Fig. 4a: *Iso*), as reported previously in other cell types[32–34], and additionally increased after application of hypotonic solution (Fig. 4a: *Hypo*), as previously observed in HeLa cells[10]. The current–voltage relationship showed that osmotic cell swelling increased the currents in the entire range of voltages between +100 and −100 mV (Fig. 4b), as clearly seen at ±100 mV (Fig. 4c). Finally, the hypotonicity-augmented cationic currents were suppressed by application of extracellular $Mg^{2+}$, and $Mg^{2+}$-induced suppression of inward currents was more prominent than that of the outward currents (Fig. 5), as previously observed in HeLa cells[10]. Such voltage-dependent sensitivity of the currents in DT40 cells to extracellular $Mg^{2+}$ administration further confirmed that these are TRPM7-mediated currents. Therefore, endogenous VSOR and TRPM7 channel activities are present in DT40 cells.

**Functional linkage between TRPM7 and VSOR upon osmotic cell swelling in DT40 cells.** Next, we examined the possibility of a functional linkage between TRPM7 and VSOR activities in DT40 cells. Our experimental approach involved simultaneous measurements of cationic TRPM7 and anionic VSOR currents by clamping, in an alternating manner, the membrane potentials

close to the equilibrium potentials for VSOR-conductive anions ($Cl^-$ plus aspartate$^-$) and TRPM7-conductive cations ($Cs^+$ plus $Na^+$), respectively. Since there is no reported data on the permeability ratio between $Cl^-$ and aspartate$^-$ ($P_{asp}/P_{Cl}$) through the VSOR anion channel and that between $Cs^+$ and $Na^+$ ($P_{Na}/P_{Cs}$) through the TRPM7 cation channel endogenously expressed in DT40 cells, we first determined these values by measuring the shifts of the reversal potential in response to replacement of extracellular $Cl^-$ and $Cs^+$ with aspartate$^-$ and $Na^+$, respectively. We obtained a $P_{asp}/P_{Cl}$ value for VSOR of 0.053 ± 0.004 ($n = 11$) and the $P_{Na}/P_{Cs}$ value for TRPM7 of 0.76 ± 0.02 ($n = 11$). Based on these values, cationic TRPM7 currents and anionic VSOR currents were simultaneously measured at the equilibrium potentials for anions and cations, −30 and +30 mV, respectively, while at ramp-clamp, as shown in Fig. 6. The cationic current was predominantly activated within 2 min immediately after application of hypotonic solution (Fig. 6a: filled circles) with shifting the reversal potential ($E_{rev}$) to the positive direction (open triangles) up to the mean peak (positive) value of 6.7 ± 0.6 mV ($n = 7$). On the other hand, thereafter, the anion current became predominantly activated (Fig. 6a: open circles) with shifting the $E_{rev}$ value to the negative direction (open triangles) down to the mean bottom (negative) value of −14.7 ± 0.7 mV ($n = 7$: $P$ (=0.000000000047) <0.0001 compared to the above mean peak

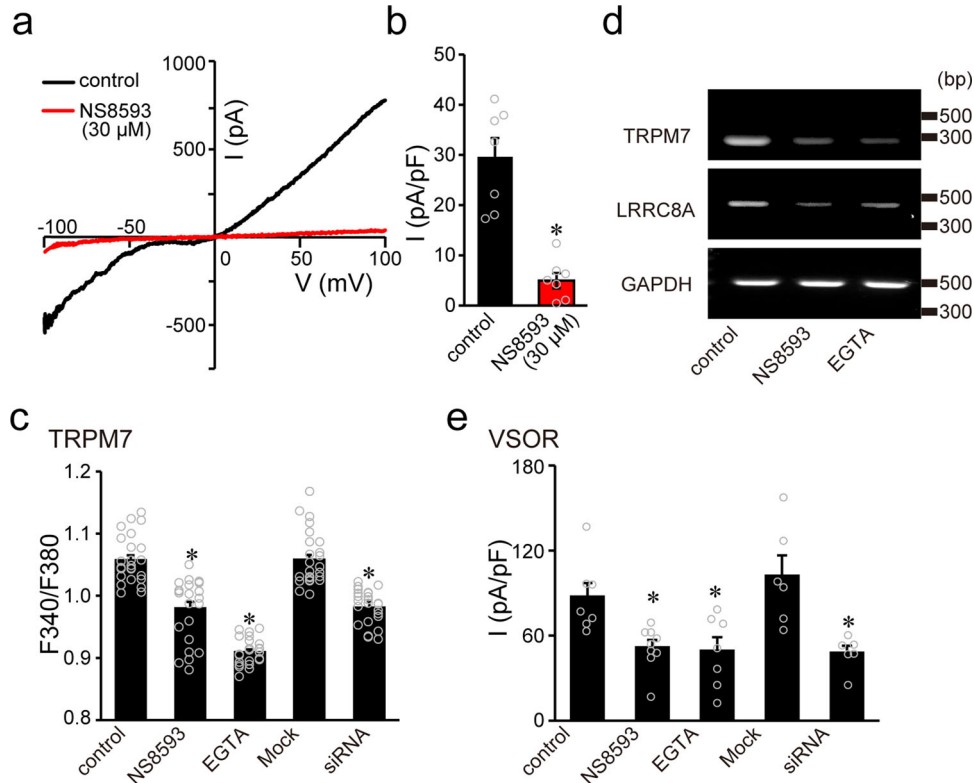

**Fig. 2 TRPM7-mediated steady-state Ca$^{2+}$ influx regulates the mRNA expression levels of TRPM7 and LRRC8A through the regulation of intracellular Ca$^{2+}$ concentration in HeLa cells. a** Representative $I–V$ relationships of currents in response to ramp pulses (50 ms duration) from $-100$ to $+100$ mV before (*control*) and after 2-day treatment with 30 μM NS8593 (*NS8593*). **b** Mean peak current densities at $+100$ mV in the absence (black column) and presence of NS8593 (red column) ($n = 6$). * indicates a $P$ ($=0.000058$) value of $<0.0001$ compared to the corresponding control (left column) by $t$-test. **c** Cytosolic Ca$^{2+}$ levels were determined by fura-2 ratio imaging in the cells before (*control*) and after treatment with 30 μM NS8593 for 2 days (*NS8593*), 2 mM EGTA for 2 days (*EGTA*), mock siRNA for 2–3 days (*Mock*), and TRPM7-siRNA for 2–3 days (*siRNA*) ($n = 24$). *$P$ ($=0.0000000014$, $0.000000000067$, and $0.0000000032$ for *NS8593*, *EGTA*, and *siRNA*, respectively) $<0.0001$ compared to the corresponding control data by one-way ANOVA followed by the post hoc Tukey's test. **d** Representative PCR products of human TRPM7, LRRC8A, and GAPDH in the cells before (*control*) and after 2-day treatment with 30 μM NS8593 (*NS8593*) or 2 mM EGTA (*EGTA*) from three independent experiments. The expression of TRPM7 and LRRC8A mRNAs was greatly decreased in NS8593- or EGTA-treated cells, while that of GAPDH mRNA stayed almost constant. The nucleotide sequences of the PCR products obtained with TRPM7- and LRRC8A-specific primers were completely identical to the corresponding sequences in TRPM7 (human: NM_017672) and LRRC8A (human: NM_001127244), respectively. **e** Swelling-induced peak VSOR current densities recorded at $+40$ mV before (*control*) and after treatment with NS8593 (*NS8593*), EGTA (*EGTA*), negative control siRNA (*Mock*), and TRPM7-siRNA (*siRNA*) ($n = 6–9$). Each column represents the mean ± SEM (vertical bar) of $n$ samples. *$P$ ($=0.038$, $0.036$, and $0.037$ for *NS8593*, *EGTA*, and *siRNA*, respectively) $<0.05$ compared to corresponding control data by one-way ANOVA followed by the post hoc Tukey's test.

value by $t$-test). Here, as seen in Fig. 6a ($I–V$ curves), the $I–V$ relationship at the time point (i) corresponds almost solely to that of TRPM7 currents (see Fig. 4b). As time passed from time point (ii) to time point (iii), the VSOR-like $I–V$ relationship (see Fig. 3b) became predominant (Fig. 6a: right panel). When peak current densities were plotted, the relationship of TRPM7-mediated cationic current measured at $-30$ mV and VSOR-mediated anionic current measured at $+30$ mV for each time point over 8 min after hypotonic stimulation in the same cell could be fitted well with a straight line of $y = -5.6x - 2.0$ ($R^2 = 0.97$), as shown in Fig. 6b, showing that there is indeed a functional correlation between VSOR and TRPM7 activity in DT40 cells.

**Effects of TRPM7 expression on LRRC8A expression and RVD in DT40 cells.** As shown earlier for HeLa cells (Fig. 1), down-regulation of LRRC8A expression in the cells treated with TRPM7-siRNA suggests that a molecular relationship between TRPM7 and LRRC8A could be, at least in part, responsible for the observed functional coupling between TRPM7 and VSOR activity in HeLa cells. To pursue a more in-depth investigation into the molecular basis of this functional coupling, we used chicken

DT40 cells which are more amenable to genetic manipulation. As shown in Fig. 7a, b and Supplementary Fig. 3c, when gallus TRPM7 (gTRPM7) was knocked out in DT40 cells (*KO*), LRRC8A mRNA expression was largely suppressed. On the other hand, when gTRPM7-deficient DT40 cells were complemented with human TRPM7 (hTRPM7), the mRNA expression level of gallus LRRC8A was prominently increased (Fig. 7a, b: +*hM7* and Supplementary Fig. 3c: +*hM7*). As plotted in Fig. 6c, gTRPM7-deficient DT40 cells complemented with wild-type (WT) hTRPM7 exhibited a functional correlation with a linear relationship, fitted with $y = -8.6x - 16.2$ ($R^2 = 0.83$), between VSOR activity and TRPM7 activity, as was the case in control DT40 cells (Fig. 6b). These results indicate that both LRRC8A expression and VSOR activity are closely correlated with TRPM7 expression.

Since VSOR activity plays an essential role in cell volume regulation, RVD, after osmotic cell swelling by serving as the volume-regulatory Cl$^-$ efflux pathway (see Reviews:[12,20,35]), we next examined the effects of knockout of gTRPM7 and complementary expression with hTRPM7 on RVD in DT40 cells. As shown in Fig. 7 (c, e), control DT40 cells responded to hypotonic stimulation (78% osmolarity) with RVD within 30 min after rapid

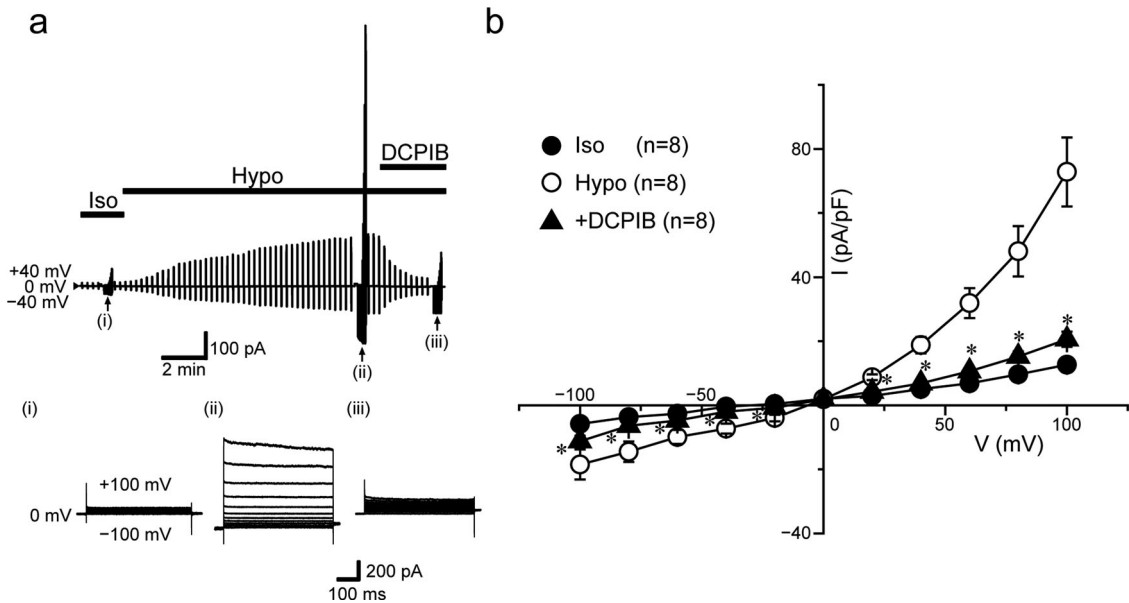

**Fig. 3 Endogenous VSOR activity in DT40 cells. a** Representative recording showing the time course of whole-cell currents before (*Iso*) and after (*Hypo*) hypotonic stimulation (86% osmolality) in the absence and presence of DCPIB (10 µM). Currents were elicited by alternating pulses from 0 to ±40 mV. Step pulses from −100 to +100 mV in 20-mV increments were applied at time points indicated by (i), (ii), and (iii). Bars above the recordings indicate the time during applications of hypotonic solution and DCPIB (10 µM). **b** Mean *I–V* relationships for isotonic (*Iso*) and hypotonicity-induced (*Hypo*) whole-cell currents before and during application of DCPIB (+*DCPIB*: n = 8). Each data point represents the mean ± SEM (vertical bar) of *n* samples. *P (=0.048, 0.028, 0.0499, 0.0054, 0.016, 0.0018, 0.00050, 0.00056, 0.0011, and 0.00031 at −100, −80, −60, −40, −20, +20, +40, +60, +80, and +100 mV, respectively) <0.05 compared to corresponding data in *Hypo* group by *t*-test.

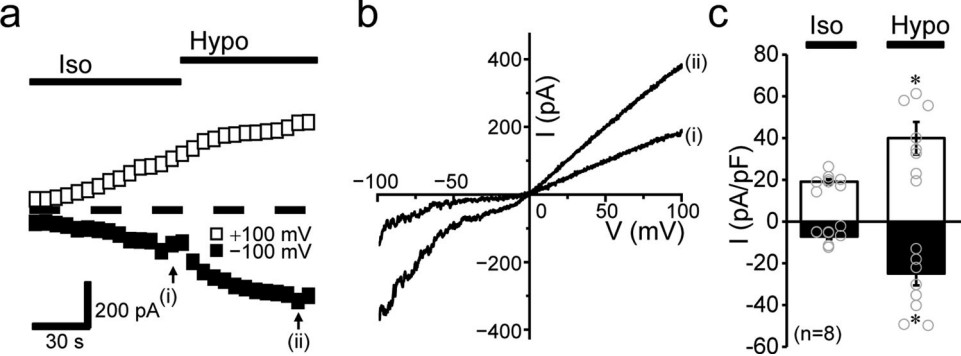

**Fig. 4 Augmentation of whole-cell TRPM7 currents by hypotonic stimulation in DT40 cells. a** Effects of hypotonic stimulation on whole-cell TRPM7 currents under ramp-clamp (every 5 s at 4 mV/ms) in the cells equilibrated with the ATP-free and low-Cl⁻ intracellular (pipette) solution. Representative peak outward and inward currents recorded at +100 and −100 mV, respectively, before (*Iso*) and after (*Hypo*) application of hypotonic (86% osmolarity) solution. **b** Representative *I–V* relationships of currents in response to ramp pulses (50-ms duration) from −100 to +100 mV before (at i in **a**) and during (at ii in **a**) hypotonic stimulation. **c** Peak current densities at +100 (blank columns) and −100 mV (filled columns) in isotonic (*Iso*) and hypotonic (*Hypo*) conditions (n = 8). Each data point represents the mean ± SEM (vertical bar) of *n* samples. *P (=0.0029 and 0.00017 at +100 and −100 mV, respectively) <0.005 compared to corresponding data in *Iso* group by *t*-test.

osmotic cell swelling (control); this RVD event was mostly abolished by a VSOR blocker DCPIB (+*DCPIB*). The RVD event was also largely inhibited in gTRPM7-deficient DT40 cells (*KO*), whereas it was significantly augmented in gTRPM7-deficient DT40 cells complemented with WT hTRPM7, as shown in Fig. 7 (d, e: +*hM7*). Taken together, these experiments show that TRPM7 expression upregulates the molecular expression of LRRC8A, thereby increasing VSOR activity and enhancing RVD efficacy.

**Roles of the kinase domain and its enzyme activity of TRPM7 in VSOR regulation in DT40 cells.** Since TRPM7 possesses both ion-conductive and enzymatic activities[36], we next assessed whether its kinase domain or its kinase activity is involved in

TRPM7-mediated VSOR regulation. We used gTRPM7-deficient DT40 cells (*KO*) and those complemented with (i) WT hTRPM7 (+*WT*), (ii) the Δ-kinase construct in which the entire α-kinase domain of hTRPM7 was deleted following amino acid 1569[37] (+*Δ-kinase*), and (iii) the K1648R construct in which the α-kinase was rendered inactive with a point mutation in the ATP binding site[37,38] (+*K1648R*). First, we confirmed the presence of expressed hTRPM7 channels in the plasma membrane of gTRPM7-deficient DT40 cells by immunostaining using an antibody to the HA tag of hTRPM7 and viewing Alexa-488 fluorescence under a confocal laser microscope. Indeed, as shown in Fig. 8a, there is robust positive staining in the periphery region of gTRPM7-deficient DT40 cells complemented with WT hTRPM7 (+*WT*). gTRPM7-deficient DT40 cells complemented with Δ-kinase

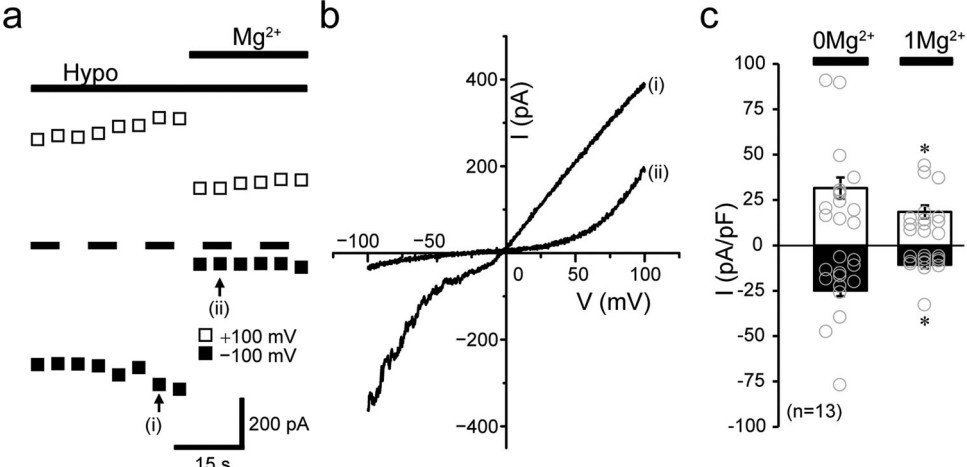

**Fig. 5 Mg²⁺ sensitivity of TRPM7 currents in DT40 cells. a** Representative recording showing the time course of whole-cell currents recorded at +100 and −100 mV during hypotonic stimulation in the cells before and after extracellular application of 1 mM Mg²⁺ (Mg²⁺). Currents were elicited by ramp-clamp (every 5 s at 4 mV/ms). Bars above the recording indicate the time during applications of hypotonic solution (Hypo) and Mg²⁺. **b** Representative I–V relationships of currents in response to ramp pulses from −100 to +100 mV before (at i in **a**) and during application of Mg²⁺ (at ii in **a**). **c** Peak current densities at +100 (blank columns) and –100 mV (filled columns) before (0Mg²⁺) and after application of 1 mM Mg²⁺ (1Mg²⁺) (n = 13). Each data point represents the mean ± SEM (vertical bar) of n samples. *P (=0.048 and 0.021 at +100 and −100 mV, respectively) <0.05 compared to corresponding data in 0Mg²⁺ group by t-test.

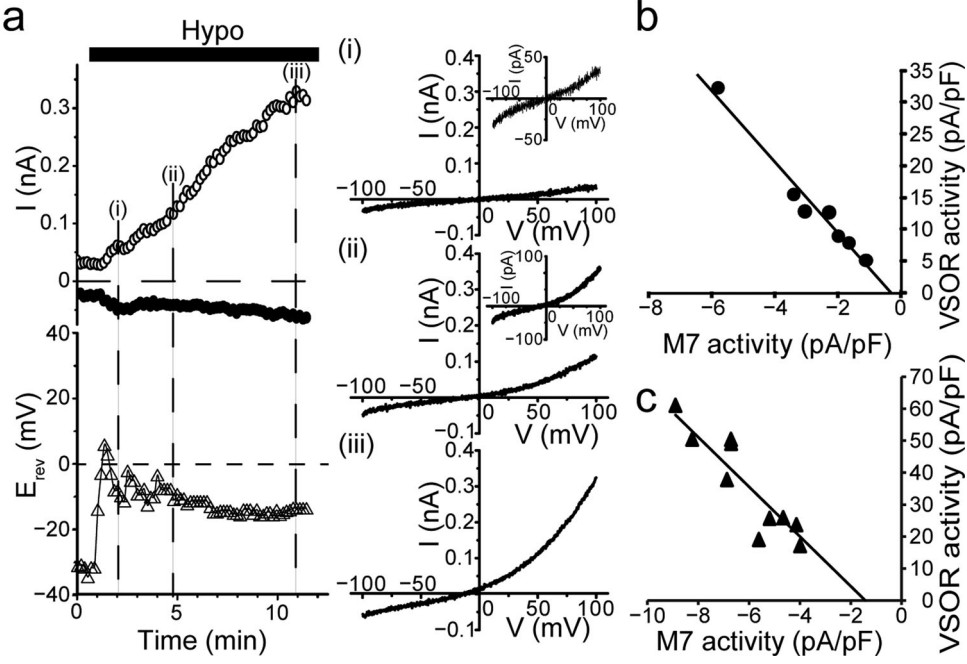

**Fig. 6 Real-time simultaneous recordings of TRPM7 and VSOR activity after hypotonic stimulation in DT40 cells. a** Whole-cell current response to hypotonic stimulation. Upper left panel shows representative recording exhibiting the time courses of whole-cell currents observed at +30 mV (open circles) and –30 mV (filled circles), which correspond to the equilibrium potentials for cations and anions, respectively, under the ionic conditions employed for this recording. Lower left panel shows the reversal potentials ($E_{rev}$: open triangles) after application of hypotonic solution (Hypo) in WT DT40 cells. Currents were elicited by ramp-clamp (every 10 s at 4 mV/ms). The value of $E_{rev}$ was evaluated by analyzing the measured reversal potentials of the all current traces in response to ramp pulses. Right panels show representative I–V relationships of currents in response to ramp pulses from –100 to +100 mV applied at (i), at (ii), and at (iii) in the left panel. Insets represent I–V curves expanded in the y-axis direction. **b, c** Relationships between the mean peak current densities recorded at +30 mV (VSOR activity) and those at –30 mV (M7 activity) (n = 7) in WT DT40 cells (**b**) and in gTRPM7-deficient DT40 cells complemented with hTRPM7 (**c**).

(+Δ-kinase) and K1648R (+K1648R) also showed similar staining patterns, although fewer +Δ-kinase cells were positively stained (Fig. 8b). By contrast, when gTRPM7 was knocked out, no Alexa-488 fluorescence could be seen in cells (Fig. 8a: KO). Next, western blot analysis using an antibody against the HA tag

of hTRPM7 confirmed the hTRPM7 channel protein expression in DT40 cells. In fact, as shown in Fig. 8c and Supplementary Fig. 4a, gTRPM7-deficinet DT40 cells complemented with WT TRPM7 or the K1648R mutant also showed bands of similar size (+WT, +K1648R), but gTRPM7-deficient DT40 cells

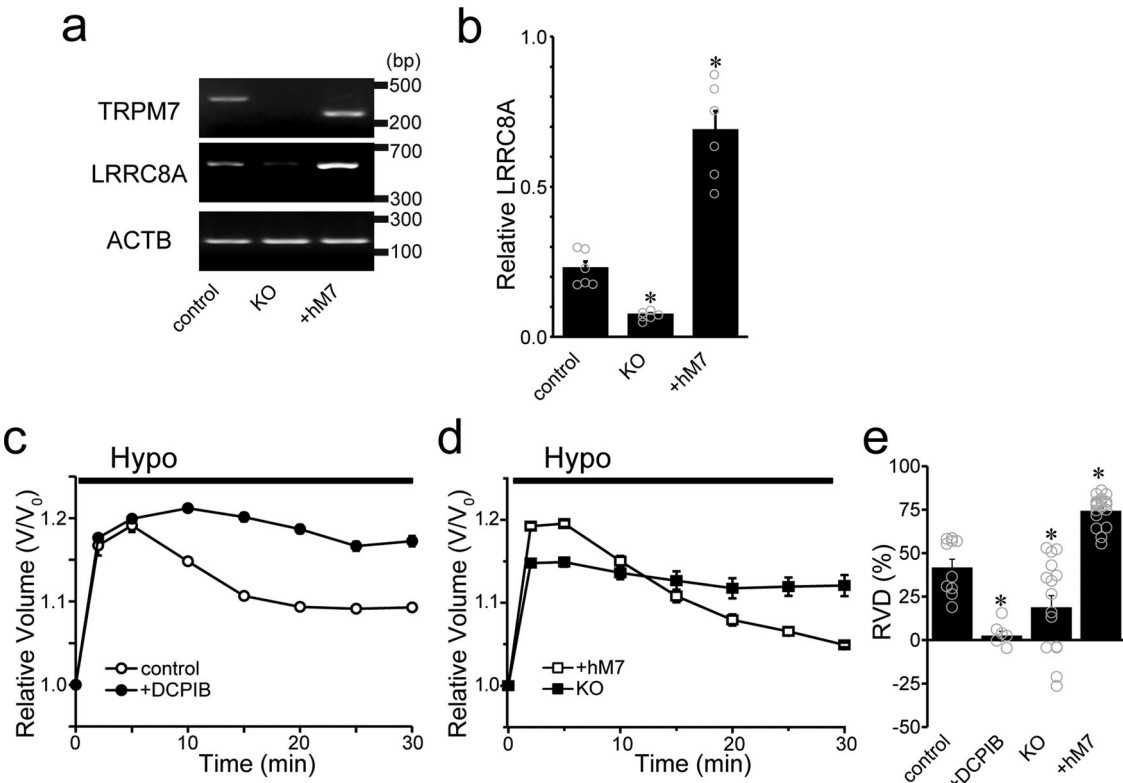

**Fig. 7 Effects of gTRPM7 knockout and complementary expression of hTRPM7 on gLRRC8A expression and RVD efficacy in DT40 cells. a** PCR products of human or chicken TRPM7, chicken LRRC8A, and the constitutively transcribed control, chicken beta-actin (*ACTB*) in control, gTRPM7-KO DT40 cells (*KO*) and hTRPM7-expressing KO cells (+*hM7*). The expression of chicken LRRC8A mRNA was greatly decreased in *KO* cells and increased in +*hM7* cells, while that of *ACTB* stayed almost constant. The nucleotide sequences of the PCR products obtained with TRPM7- and LRRC8A-specific primers were completely identical to the corresponding sequences in TRPM7 (human: NM_017672; chicken: NM_001177555) and LRRC8A (chicken: XM_015279569.2), respectively. **b** The bar graphs represent the relative expression values of the optical densities in pixels of the PCR bands of gLRRC8A compared to the corresponding band of *ACTB* control. These values were calculated from six independent PCR amplifications. *$P$ (=0.039 and 0.0000021 for *KO* and +*hM7*, respectively) <0.05 compared to control data by one-way ANOVA followed by the post hoc Tukey's test. **c, d** Time courses of changes in the mean cell volume of WT DT40 cells after a hypotonic challenge (*Hypo*: 78% osmolarity). The RVD event was inhibited by DCPIB (10 μM) (**c**) and gTRPM7-KO (*KO*), whereas it was facilitated in hTRPM7-expressing *KO* cells (+*hM7*) (**d**). **e** Summarized data showing the percent volume recovery (RVD) at 30 min after application of hypotonic solution (*n* = 6–20). Each data point represents the mean ± SEM (vertical bar) of *n* samples. *$P$ (=0.00035, 0.011, and 0.000070 for +*DCPIB*, *KO*, and +*hM7*, respectively) <0.05 compared to corresponding data in control data by one-way ANOVA followed by the post hoc Tukey's test.

complemented with Δ-kinase were slightly smaller in size due to a *C*-terminal defect (+*Δ-kinase*). In contrast, no protein was detected in the cells in which endogenous gTRPM7 was knocked out (Fig. 8c: *KO* and Supplementary Fig. 4a).

Electrophysiological recordings of VSOR currents are shown in Fig. 8d, e and Supplementary Fig. 2a (+*M7*). In gTRPM7-deficient cells, surprisingly, VSOR currents were almost abolished (*KO*). When these *KO* cells were stably transfected with WT hTRPM7, however, VSOR currents were rescued (+*WT*). In contrast, transfection with hTRPM4 failed to rescue VSOR activity in *KO* cells, as shown in Supplementary Fig. 2a (+*M4*). In WT hTRPM7-expressing cells, the amplitude of whole-cell VSOR currents (270.4 ± 28.6 pA/pF; Fig. 8e: +*WT*) was much larger than that of VSOR currents recorded in control DT40 cells (see open circle in Fig. 3b at +100 mV; 72.8 ± 10.7 pA/pF, $P$ (=0.000095) <0.0001 by *t*-test). In Δ-kinase-expressing cells (+*Δ-kinase*), VSOR currents similar in magnitude to those in WT hTRPM7-expressing cells (+*WT*) were recorded in five of seventeen tested cells (29.4%), but could hardly be recorded in the other twelve cells (Fig. 8d: +*Δ-kinase*), as was the case of gTRPM7-deficient cells (*KO*). Thus, the averaged VSOR current in Δ-kinase-expressing cells was very much smaller than that in WT hTRPM7-expressing

cells (Fig. 8e). In contrast, in K1648R-expressing cells, swelling-induced VSOR currents, similar in magnitude to those in WT hTRPM7-expressing cells, were amply recorded (Fig. 8d, e), although their α-kinase is inactive. The whole-cell TRPM7 currents recorded in these DT40 cells under hypotonic conditions are shown in Fig. 8f, g. In all K1648R-expressing cells (+*K1648R*), the TRPM7 current amplitude was not significantly different from that in WT hTRPM7-expressing cells (+*WT*). As expected, no TRPM7-like current was observed in *KO* cells (Fig. 8f, g). In K1648R-expressing cells, the TRPM7 current amplitude was not significantly different from that in WT hTRPM7-expressing cells (Fig. 8f, g). In Δ-kinase-expressing cells, as in the case of VSOR currents, almost no TRPM7 current was observed in most cells (Fig. 8f: +*Δ-kinase*), although five of seventeen tested cells (29.4%) exhibited TRPM7 currents comparable in magnitude to those in WT hTRPM7-expressing cells, and thus the averaged TRPM7 current in Δ-kinase-expressing cells was very much smaller than that in WT hTRPM7-expressing cells (Fig. 8g: +*Δ-kinase*). These results suggest that TRPM7-mediated regulation of VSOR activity is independent of kinase activity of TRPM7 but is dependent on the presence of its cytosolic α-kinase domain, but not its enzyme activity.

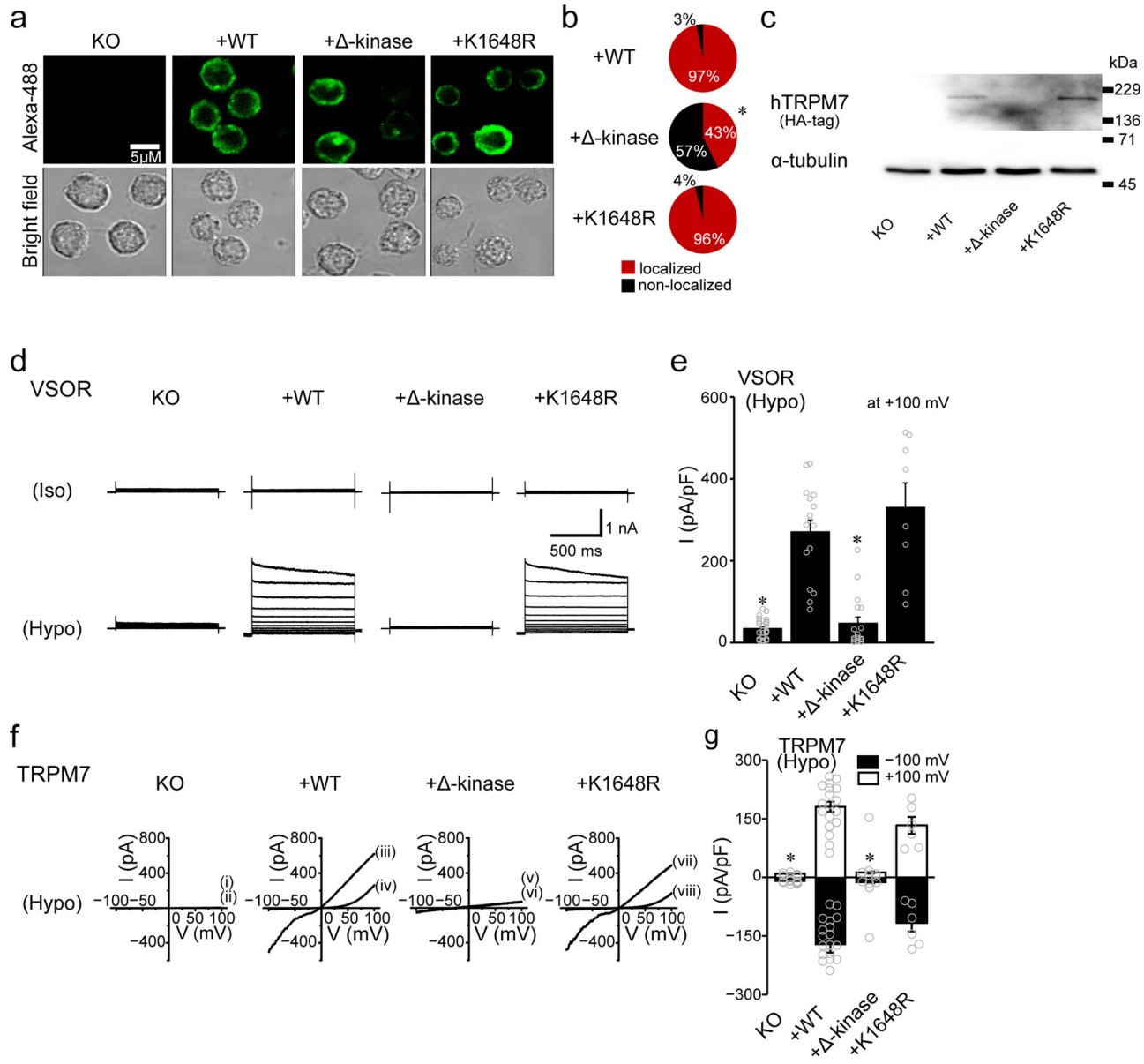

**Fig. 8 Effects of gTRPM7 knockout and complementary expression with hTRPM7 or its mutant on the protein expression monitored by immunohistochemistry as well as on VSOR and TRPM7 currents recorded by whole-cell patch-clamp in DT40 cells. a** Representative fluorescence micrographs of gTRPM7-KO DT40 cells immunostained with anti-HA antibody before (*KO*) and after complementary expression with WT hTRPM7 or its mutant (Alexa-488: green) as well as their bright field micrographs. Scale bar, 5 μm. **b** The percentage of TRPM7 localization to the plasma membrane is significantly lower in the *C*-terminal defects (+Δ-kinase: *$P$ (=1.1 × 10$^{-233}$) <0.0001 compared to +*WT data* by $\chi^2$ test). Observations and analyses were performed from 53–110 individual cells. **c** Western blot analyses of α-tubulin, HA-tagged hTRPM7 from KO, +*WT*, +Δ-*kinase*, and +*K1648R* DT40 cells. The data are representative of three independent experiments. **d** Representative whole-cell VSOR currents elicited by step pulses in isotonic conditions (*Iso*) and in hypotonic conditions (*Hypo*) in gTRPM7-knockout (*KO*) DT40 cells and those complemented with hTRPM7 (+*WT*), hTRPM7-Δ-kinase (+Δ-*kinase*), and hTRPM7-K1648R (+*K1648R*). **e** Peak VSOR current densities recorded at +100 mV (*n* = 8–26). Each column represents the mean ± SEM (vertical bar) of *n* samples. *$P$ (=0.000000000069 and 0.0000000029 for *KO* and +Δ-*kinase*, respectively) <0.0001 compared to corresponding +*WT* data by one-way ANOVA followed by the post hoc Tukey's test. **f** Representative *I*–*V* relationships of whole-cell TRPM7 currents in *KO*, +*WT*, +Δ-*kinase*, and +*K1648R* DT40 cells exposed to hypotonic (86% osmolarity) solution. The *I*–*V* curves were recorded under ramp-clamp (50-ms duration applied every 5 s at 4 mV/ms) from −100 to +100 mV in the absence (i, iii, v, vii) and presence (ii, iv, vi, viii) of 1-mM Mg$^{2+}$. **g** Peak current densities recorded at +100 mV (blank columns) and −100 mV (filled columns) in hypotonic conditions (*n* = 6–19). Each column represents the mean ± SEM (vertical bar) of *n* samples. *$P$ (=0.0000029 and 0.0012 at −100 mV and 0.00000000020 and 0.0000015 at +100 mV for *KO* and +Δ-*kinase*, respectively) <0.005 compared to corresponding data in +*WT* cells by one-way ANOVA followed by the post hoc Tukey's test.

Next, we examined possible spatial co-localization of TRPM7 and LRRC8A in HeLa cells transfected with TRPM7-EGFP and LRRC8A-mCherry. Only after a hypotonic challenge, as shown in Fig. 9 (a, b: *LRRC8A-mCherry*), LRRC8A was found to become localized to the periphery region of cells transfected with WT TRPM7 or the K1648R mutant, whereas such periphery localization of LRRC8A was not much observed in those transfected with the Δ-kinase mutant. Also, their merged images suggest that LRRC8A becomes apparently co-localized with WT TRPM7 or the K1648R mutant but not much with the Δ-kinase

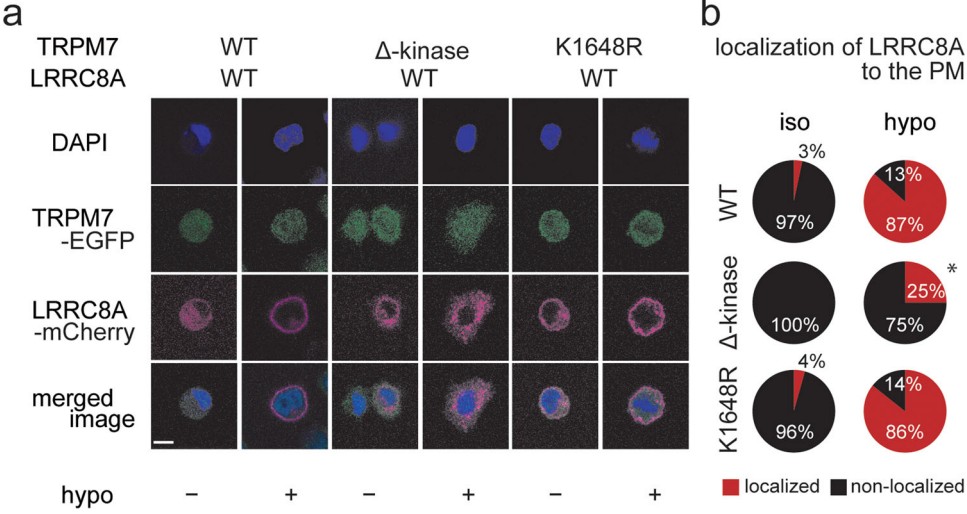

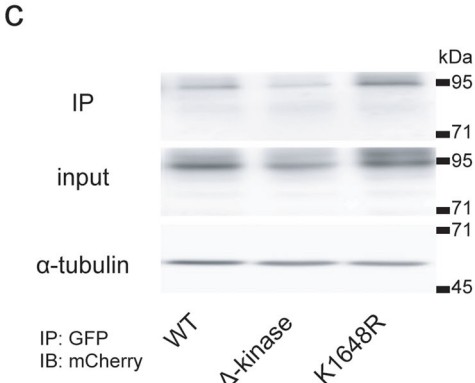

**Fig. 9 Subcellular co-localization and molecular interactions of TRPM7-WT, -Δ-kinase, and -K1648R mutants with LRRC8A after exposure to hypotonic solution. a** Confocal images of DAPI-stained HeLa cells before (*DAPI*) and after immunostaining with anti-GFP antibody for WT hTRPM7 or its mutant (*TRPM7-EGFP*: green), and those with anti-mCherry antibody for LRRC8A (*LRRC8A-mCherry*: red) during exposure to hypotonic solution (78% osmolarity). Scale bar, 5 μm. **b** The percentage of LRRC8A localization to the plasma membrane (*PM*) is significantly lower for the TRPM7 C-terminal defect (+Δ-kinase, hypo: *$P$ (=0.00060) <0.001 compared to corresponding *WT* data by $\chi^2$ test). Observations and analyses were performed from 14 to 121 individual cells. **c** Co-immunoprecipitation of hTRPM7-EGFP (*WT*), -Δ-kinase-EGFP, or -K1648R-EGFP with LRRC8A-mCherry in HEK293T cells co-transfected with hLRRC8A-mCherry and EGFP-tagged hTRPM7 or its mutant. Immunoprecipitations (*IP*) with a GFP-specific antibody were subjected to western blot with an antibody to mCherry. For input, aliquots of sample are loaded on a separate gel. The data are representative of three independent experiments.

mutant (Fig. 9a: *merged image*). Then, a possible physical interaction of TRPM7 and LRRC8A was tested by co-immunoprecipitation of LRRC8A and TRPM7 extracted from osmotically swollen HEK293T cells co-transfected with LRRC8A-mChery and TRPM7-EGFP. As shown in Fig. 9c (*WT*; Supplementary Fig. 4b: *WT*) and Supplementary Fig. 2b (*TRPM7-EGFP*), LRRC8A was found to be co-precipitated with WT TRPM7 (*IP*). In contrast, such co-precipitated bands (*IP*) were not detected when co-transfected with LRRC8A-mCherry and TRPM4-EGFP (Supplementary Fig. 2b: *TRPM4-EGFP*). The K1648R mutant of hTRPM7 exhibited similar co-precipitation with LRRC8A (*IP: K1648R*), whereas co-precipitated bands were not much found with the Δ-kinase mutant (*IP: Δ-kinase*). These data strongly suggest that LRRC8A becomes localized at the plasma membrane and interacts with TRPM7 presumably via the *C*-terminal kinase domain of TRPM7 after osmotic cell swelling.

### Discussion
VSOR and TRPM7 are ubiquitously expressed anion and cation channels, respectively, that have important roles in cell volume

regulation, viability, proliferation, and death (see Reviews:[28,39–43]). Here, we provide molecular and electrophysiological evidence that by influencing LRRC8A expression and VSOR activity, TRPM7 plays a crucial prerequisite role in RVD. We also demonstrated, for the first time, that chicken B cell-derived DT40 cells endogenously express VSOR activity (Fig. 3) and its core molecule LRRC8A (Fig. 7a and Supplementary Fig. 3c, b), and thereby can attain the RVD event after exposure to hypotonic solution (Fig. 7c, e). The whole-cell VSOR currents activated upon osmotic cell swelling exhibited outward rectification, voltage-dependent inactivation kinetics, and limited but significant permeability to aspartate$^-$ ($P_{asp}/P_{Cl}$ ~0.05). DT40 cells were also found to molecularly express TRPM7 (Fig. 7a and Supplementary Fig. 3c), as originally shown by Nadler et al.[32]. The whole-cell TRPM7 currents were observed to be activated by removal of extracellular divalent cations, augmented by osmotic cell swelling (Fig. 4), and blocked by $Mg^{2+}$ added to the extracellular solution (Fig. 5), as observed previously in other cell types[10,24,32,44]. For the first time in DT40 cells, here, we determined the $P_{Na}/P_{Cs}$ value of 0.76, the value of which is slightly smaller than that observed in mouse

Chinese hamster ovary CHO cells expressing TRPM7[36]. Finally, we show that the presence of TRPM7 kinase domain, but not its kinase activity, is required for VSOR activity (Fig. 8).

The functional coupling between VSOR and TRPM7 was shown through four lines of evidence: (i) gene-silencing knock-down of TRPM7 largely suppressed swelling-induced activation of VSOR currents in HeLa cells (Fig. 1). (ii) Pharmacological inhibition of TRPM7 activity and of TRPM7-mediated steady-state $Ca^{2+}$ influx largely reduced VSOR currents in HeLa cells (Fig. 2). (iii) In DT40 cells, TRPM7 knockout markedly abolished VSOR activity, whereas complementary expression with hTRPM7 in gTRPM7-deficient cells enhanced VSOR activity (Fig. 8d, e). (iv) The VSOR current amplitude is proportionally positively correlated to the TRPM7 current amplitude in WT DT40 cells and gTRPM7-deficient DT40 cells complemented with hTRPM7 (Fig. 6b, c). Our data also showed that the functional link between TRPM7 and VSOR is underpinned, in part, by the molecular interactions between TRPM7 and the VSOR core molecule LRRC8A based on the following observations: (i) TRPM7 gene silencing reduced expression not only of TRPM7 mRNA but also of LRRC8A mRNA in HeLa cells (Fig. 1c and Supplementary Fig. 3a). (ii) TRPM7 knockout and its complementary expression reduced and increased, respectively, expression of LRRC8A mRNA in DT40 cells (Fig. 7a and Supplementary Fig. 3c, b). (iii) Complementary expression with the Δ-kinase construct of TRPM7 very markedly suppressed not only TRPM7 currents but also VSOR currents in DT40 cells (Fig. 8d–g). (iv) Co-expression of TRPM7 Δ-kinase and LRRC8A suppressed plasmalemmal expression of LRRC8A in HeLa cells under hypotonic stimulation (Fig. 9a, b). (v) Co-expression of TRPM7 Δ-kinase and LRRC8A weakened the physical interaction between TRPM7 and LRRC8A (Fig. 9c and Supplementary Fig. 4b). Based on these pieces of cumulative evidence, we here conclude that TRPM7 serves as an essential regulator of VSOR activity and LRRC8A expression via its C-terminal α-kinase domain. Since the K1648R mutant failed to affect VSOR activity in DT40 cells (Fig. 8d, e), the plasmalemmal LRRC8A expression in HeLa cells (Fig. 9a, b), and the physical interaction between TRPM7 and LRRC8A in HEK293T cells (Fig. 9c and Supplementary Fig. 4b), it is concluded that the enzyme activity of α-kinase domain is not involved in the regulatory role of TRPM7 on VSOR expression and activity.

Functional interplays between TRP cation channels and $Ca^{2+}$-activated chloride channels were so far reported to be mediated by $Ca^{2+}$ influx via a number of TRP cation channels, such as TRPC2[45], TRPC6[46], and TRPV6[47]. Molecular or physical interaction between TRPs and TMEM16A/ANO1 was also found by Tominaga's group for TRPV4[48,49] and TRPV1[50]. On the other hand, overexpression of TRPV4 was found to induce down-regulation of VSOR activity in HEK293 cells[51]. The present study adds to this growing theme of functional and molecular interactions between TRP cation channels and anion channels in regulating important aspects of cell physiology such as RVD.

Osmotic cell swelling has been shown to activate many types of TRP cation channels, including TRPV1[51,52], TRPV2[53], TRPV4[54–63], TRPC1[64,65], TRPC5[66], TRPC6[67,68], TRPA1[69], TRPP3[70], TRPM3[71], and TRPM7[10,24]. Among them, TRPV4[72–74] and TRPC1[75] have been shown to be involved in RVD facilitation. Using the gene-silencing technique, we previously provided molecular evidence for an involvement of swelling-induced activation of TRPM7 in RVD in HeLa cells[10]. Here, molecular evidence for this involvement was supplemented by observations that TRPM7 gene knockout and complementary expression with hTRPM7 effectively abolished and augmented, respectively, the RVD event in DT40 cells (Fig. 7d, e).

In sum, we conclude that TRPM7 dually contributes to RVD not only by stimulating $Ca^{2+}$-activated $K^+$ channels but also by augmenting volume-sensitive $Cl^-$ channels upon osmotic cell swelling. The former occurs when swelling-induced activation of TRPM7 causes $Ca^{2+}$ influx, which leads to $Ca^{2+}$-induced $Ca^{2+}$ release from intracellular $Ca^{2+}$ stores[76], thereby bringing about a sustained rise of intracellular-free $Ca^{2+}$ level and eventually activating IK1 $Ca^{2+}$-activated $K^+$ channels (see Review:[43]). The latter occurs when swelling-induced TRPM7 activation augments both functional activities of VSOR $Cl^-$ channels and molecular expression of the core component of VSOR, LRRC8A. We here showed, for the first time, that TRPM7 dually exerts an augmenting action on VSOR activity: first by molecularly increasing expression of LRRC8A mRNA through the mediation of steady-state $Ca^{2+}$ influx and second by stabilizing the plasmalemmal expression of LRRC8A protein through the molecular interaction with the C-terminal α-kinase domain in a manner independent of its enzyme activity.

Our findings have important implications because of the physiological and pathological roles of osmotic swelling, VSOR activity, and TRPM7 channels. Depending on cell and tissue types, these physiological roles could include immunological processes, insulin secretion, adipocyte insulin signaling, astrocyte-neuron communication, etc. Dysfunction of RVD is known to be closely coupled to, and implicated in apoptotic, necrotic, and ischemic cell death processes under physiological and pathological conditions (see Reviews:[2,43,77]). Therefore, the molecular and functional link between TRPM7 and VSOR could an efficient target for therapeutic development and clinical treatments.

## Methods

**Cell culture**. Human cervical HeLa cells or human embryonic kidney 293T (HEK293T) cells were grown as a monolayer in modified Eagle's medium or Dulbecco's modified Eagle's medium supplemented with 10% fetal bovine serum, 40 U/ml penicillin G, and 100 μg/ml streptomycin under 95% air–5% $CO_2$ atmosphere at 37 °C. For electrophysiological experiments, the cells were detached from the plastic substrate and cultured in suspension with stirring for 15–300 min. For some experiments, HeLa cells were provided after culturing for 2 days in the presence of 30 μM NS8593 or 2 mM EGTA. Chicken B cell-derived DT40 cells were maintained in RPMI 1640 medium supplemented with 10% fetal bovine serum, 1% chicken serum, 100 U/ml penicillin, and 100 μg/ml streptomycin in a humidified 95% air, 5% $CO_2$ atmosphere at 37 °C.

**DT40 cell line construction**. Construction of a stable gTRPM7-deficient DT40 cell line was made as previously described[32,37]. Briefly, the targeting construct pLTRPC7 (TRPM7) -bsr was introduced into DT40 cells by targeting the second TRPM7 allele of the gTRPM7 heterozygotes. The gTRPM7-deficient cell line was obtained by culturing in a medium supplemented with 10 mM $Mg^{2+}$ and selecting with Blasticidin. Tetracycline-inducible and stable expression of hTRPM7-WT, hTRPM7-K1648R, and hTRPM7-Δ-kinase DT40 cells was made as reported previously[37]. Briefly, the pcDNA™4/TO vector was prepared to contain a sequence in which two tetracycline operator sequences ($TetO_2$) are inserted between the TATA box of the CMV promoter and the transcription initiation site. The hTRPM7-constructed pcDNA™4/TO vectors (Invitrogen) were transfected into a gTRPM7-deficient DT40 cell line and subjected to a drug selectivity test supplemented with blasticidin and zeocin.

**Construction of plasmid**. The full-length human TRPM7 cDNA[24] was deleted and replaced with cDNAs of the following two different mutants prepared by using the QuikChange Lightning Site-Directed Mutagenesis Kits (Agilent Technologies, CA, USA): the Δ-kinase mutant, in which the entire α-kinase domain following amino acid 1569 in TRPM7, was deleted[37], and the K1648R mutant, the enzyme activity of which is inactive by a point mutation at the ATP binding site in the amino acid α-kinase domain[37,38]. Full-length human LRRC8A cDNA[78], human WT TRPM7, TRPM7-Δ-kinase, TRPM7-K1648R, and human TRPM4[79] have been subcloned into pEGFP-N (Takara-Bio, Shiga, Japan) or pmCherry-N (Takara-Bio). The insertion area was confirmed by sequencing the entire insertion area. The recombinant plasmids for heterologously expression of pIRES2-EGFP-TRPM7, and pIRES2-EGFP-TRPM4 were subcloned into pIRES2-EGFP vector (Clontech, Mountain View, CA), as previously described[24,79]. Briefly, the end of full-length human TRPM7 cDNA and human TRPM4 cDNA was cleaved with an appropriate restriction enzyme and inserted into the multicloning site of pIRES2-EGFP.

**Electrophysiology**. Cells were dissociated by mechanical agitation and lodged onto coverslips placed in tissue culture dishes. Membrane currents of these cells were

recorded at room temperature (22–27 °C) using the whole-cell mode of the patch-clamp technique, with an Axopatch 200B (Axon Instruments/Molecular Devices, Union City, CA, USA) patch-clamp amplifier. For whole-cell recordings, patch electrodes prepared from borosilicate glass capillaries had an input resistance of 3–7 MΩ. Current signals were filtered at 5 kHz with a four-pole Bessel filter and digitized at 20 kHz. pCLAMP software (version 10.5.1.0; Axon Instruments/ Molecular Devices) was used for command pulse control, data acquisition, and analysis. Data were also analyzed using Origin software (OriginLab Corp., Northampton, MA, USA). For whole-cell recordings, the series resistance was compensated (to 70–80%) to minimize voltage errors, and the liquid junction potential emerged between the extracellular solution and the pipette solution was evaluated by using Clampex software (Axon Instruments/Molecular Devices) and compensated before each experiment.

For VSOR current measurements, the amplitude of whole-cell current was recorded by applying alternating pulses of ±40 mV (500-ms duration) from a holding potential of 0 mV every 15 s. To observe voltage dependence of the current profile and the inactivation kinetics at large positive potentials, step pulses (800-ms duration) were applied from a pre-potential of −100 mV to test potentials of −100 to +100 mV in 20-mV increments. The amplitude of the instantaneous current was measured at ~5 ms after the start of the step pulse. Whole-cell recordings of VSOR currents were performed using the external solution containing (in mM) 110 CsCl, 5 $MgSO_4$, and 10 HEPES (pH 7.4 adjusted with CsOH, and osmolality adjusted to 320 mosmol/kg $H_2O$ in isotonic solution or 275 mosmol/kg $H_2O$ in hypotonic solution with D-mannitol. The pipette solution contained (in mM) 110 CsCl, 2 $MgSO_4$, 1 EGTA, 10 HEPES, 2 $Na_2ATP$, and 0.3 GTP (pH 7.3 adjusted with CsOH, and osmolality adjusted to 300 mosmol/kg $H_2O$ with D-mannitol; ionic strength ~130 mM). The liquid junction potential emerged between these solutions was evaluated to be +0.5 mV. For Fig. 2e, cell swelling was induced by adjusting the intracellular osmolality to 360 mosmol/kg $H_2O$ with D-mannitol and recorded the amplitude of whole-cell currents. To evaluate the $P_{asp}/P_{Cl}$ value for VSOR currents in DT40 cells, CsCl was replaced with Cs-aspartate in the pipette solution.

For TRPM7 current recordings, ramp pulses were applied from −100 to +100 mV at a speed of 1 mV/ms. The intracellular (pipette) solution contained (in mM): 100 Cs-aspartate, 1 EGTA, 10 HEPES, and 0.5 CsCl (pH adjusted to 7.4 with CsOH, and osmolality adjusted to 300 mosmol/kg $H_2O$ with D-mannitol; ionic strength ~110 mM). Since VSOR activation is known to be induced or facilitated by reduction of intracellular ionic strength[80–85], we here eliminated ATP and most Cl⁻ ions from the intracellular solution to prevent possible VSOR Cl⁻ current contamination[2,12]. The bath solution contained (in mM): 100 Na-aspartate (or Cs-aspartate) and 10 HEPES (pH adjusted to 7.4 with NaOH (or CsOH), and osmolality adjusted to 320 mosmol/ kg $H_2O$ with D-mannitol). The liquid junction potential emerged between these solutions was evaluated to be +7.9 mV. To estimate the $P_{Na}/P_{Cs}$ value for TRPM7 currents in DT40 cells, Cs-aspartate was replaced with Na-aspartate in the pipette solution.

In the case of simultaneous measurements of cationic and anionic macroscopic currents for Fig. 6, the pipette solution was replaced with (in mM) 28.5 CsCl, 81.5 NMDG, 81.5 L-aspartic acid, 2 $MgSO_4$, 1 EGTA, 10 HEPES, 2 $Na_2ATP$, and 0.3 GTP (pH 7.4 adjusted with CsOH; and osmolality adjusted to 300 mosmol/kg $H_2O$ with D-mannitol; ionic strength ~130 mM). The external solution containing (in mM) 110 CsCl, 0.1 $MgSO_4$, 0.1 $CaSO_4$, and 10 HEPES (pH 7.4 adjusted with CsOH, and osmolality adjusted to 320 mosmol/kg $H_2O$ in isotonic solution or 275 mosmol/kg $H_2O$ in hypotonic solution with D-mannitol). The liquid junction potential emerged between these solutions was evaluated to be +0.6 mV.

**Mean cell volume measurements**. Mean cell volume was measured at room temperature by electronic sizing with a Coulter-type cell size analyzer (CDA-500; Sysmex, Hyogo, Japan). The mean volume of the cell population was calculated from the cell volume distribution measured after the machine was calibrated with latex beads of known volume. Isotonic or hypotonic solution (275 or 215 mosmol/kg $H_2O$ adjusted by D-mannitol) contained (in mM) 100 NaCl, 5 KCl, 0.4 $MgCl_2$, 0.42 $CaCl_2$, 10 D-glucose, and 5 HEPES (pH 7.4 adjusted by NaOH). Relative cell volumes in Fig. 7 are defined by the following equation: relative cell volume = $V_A/V_{Ctl}$, where $V_{Ctl}$ and $V_A$ are the mean cell volumes before and after hypotonic challenge, respectively.

**RNA isolation and RT-PCR**. Total cellular RNA was extracted from DT40 and HeLa cells by using NucleoSpin® RNA Plus (Takara-Bio) according to the protocol supplied by the manufacturer. The concentration and purity of RNA were determined using a Nanodrop-ND1000 (Thermo Fisher Scientific, Waltham, MA, USA). Total RNA samples were reverse-transcribed at 42 °C for 60 min with Prime-Script RTase using the Prime-Script™ II High Fidelity RT-PCR Kit (Takara-Bio), according to the manufacturer's protocols. Expression levels of TRPM7 and LRRC8A in the cDNA from DT40 or HeLa cells were determined by PCR. As a positive control, we amplified the partial sequence of ACTB (beta-actin) or GAPDH. PCR was done using KOD-Plus-Ver.2 (Toyobo, Osaka, Japan) under the following conditions: predenaturation at 94 °C for 2 min, followed by 25–35 cycles of denaturation at 98 °C for 10 s and annealing at 55–63 °C for 30 s, and a final extension at 68 °C for 30 s. The sequences of gene-specific primers were designed with Primer3 software (http://bioinfo.ut.ee/primer3-0.4.0/) and synthesized by FASMAC (Kanagawa, Japan) or Sigma-Aldrich (St. Louis, MO, USA); and the predicted lengths of PCR products are as follows: gACTB (beta-actin) (137 bp)

forward and reverse primers: 5′-GATATGGAGAAGATCTGGCA-3′ and 5′-GGTCTCAAACATGATCTGTGT-3′, respectively; gLRRC8A (176 bp) forward and reverse primers: 5′-TGCCATTCTGGAAGACAGCA-3′ and 5′-GGAGGTCC AAGCTGCAAATG-3′, respectively; gTRPM7 (405 bp) forward and reverse primers: 5′-TGCCATTCTGGAAGACAGCA-3′ and 5′-GGAGGTCCAAGCTGCA AATG-3′, respectively; hLRRC8A (419 bp) forward and reverse primers: 5′-ACT TCCCCTACCTGGTGCTT-3′ and 5′-AGAGGCGGTACACAATGTCC-3′, respectively. hGAPDH (496 bp) forward and reverse primers: 5′-GGTGAAGGT CGGAGTCAACG-3′ and 5′-CAAAGTTGTCATGGATGACC-3′, respectively; hTRPM7 (276 bp) forward and reverse primers: 5′-CACTTGGAAACTGGA ACC-3′ and 5′-CGGTAGATGGCCTTCTACTG-3′, respectively; hTRPM4 (274 bp) forward and reverse primers[86]: 5′-GGCGGAGACCCTGGAAGACA-3′ and 5′-CCAAGCCACAGCCAAACG-3′, respectively.

**Modification of gene expression**. To reduce the expression of human TRPM7 or human TRPM4, siRNA-mediated knockdown was performed in HeLa cells at 70–80% confluence. For siRNA transfection, Lipofectamine 2000 (Invitrogen Life Technologies, Carlsbad, CA, USA) was employed according to the manufacturer's protocols. siRNAs for hTRPM7 or hTRPM4 were designed and synthesized by Sigma-Aldrich. The target sequence and knockdown efficiency of the siRNA used were those previously confirmed[10,87]. A negative control siRNA conjugated with Alexa Fluor 488 (AllStars; Qiagen, Hilden, Germany) was used for mock cells. Cells were used in experiments 48–72 h after siRNA transfection.

Gene expression of TRPM7-WT, TRPM7-K1648R, and TRPM7-Δ-kinase in the DT40 cell line was induced by administration of 1 μg/ml tetracycline for 24 h, and its efficiency has been described previously[37]. gTRPM7-deficient DT40 cells were transfected with a recombinant plasmid pIRES2-EGFP vector (Clontech, Mountain View, CA), pIRES2-EGFP-TRPM7, or pIRES2-EGFP-TRPM4, by lipofectamine 2000 reagent. Briefly, DNA and lipofectamine 2000 were mixed in Opti-MEM™ medium (Thermo Fisher Scientific) to form a complex. The formed complex was added to a 6-cm non-coated dish containing DT40 cells and allowed to stand in a $CO_2$ incubator at 37 °C for 6 h. Then, the cells were transferred to and incubated in fresh medium for an additional 24–48 h. Transfection of LRRC8A-mCherry with pEGFP-N, TRPM4-EGFP, or TRPM7-EGFP into HEK293T or HeLa cells were also performed by using Lipofectamine 2000 for 36–48 h.

**Confocal microscopy**. gTRPM7-deficient DT40 cells stably expressing hTRPM7-WT, -K1648R, or -Δ-kinase were grown on 6-cm dishes. For the experiments of Fig. 9, hTRPM7- or K1648R-, or Δ-kinase-EGFP and hLRRC8A-mCherry were co-transfected in HeLa cells. Cells were fixed with 4% paraformaldehyde and then permeabilized with 0.1% Triton X-100. The cells were incubated with Anti-HA (1:1000 dilution, 11 867 423 001; Roche Diagnostics, Mannheim, Germany), anti-GFP (1:1000 dilution, 50430-2-AP; Proteintech, IL, USA), or anti-mCherry (1:1000 dilution, GTX630189; GeneTex, Inc., CA, USA) primary antibody followed by subsequent incubation with the anti-rat Alexa-488-conjugated (1:4000 dilution, A-11006; Thermo Fisher Scientific), anti-rabbit Alexa-488-conjugated (1:4000 dilution, A-1100; Thermo Fisher Scientific), or anti-mouse Alexa-647-conjugated (1:4000 dilution, A-21236; Thermo Fisher Scientific) secondary antibody. Fluorescence images were acquired with a confocal laser-scanning microscope (Zeiss LSM710, Carl Zeiss Microscopy GmbH, Jena, Germany), which equips a ×40 oil objective lens. Alexa-488, Alexa-647, or DAPI signals were acquired, and line analyses were made with ZEN software (Carl Zeiss).

**Intracellular $Ca^{2+}$ measurements**. The intracellular $Ca^{2+}$ concentration was measured by ratio imaging of fura-2 fluorescence using SPARK10M plate reader (Tecan Group Ltd., Männedorf, Switzerland). After culturing $1 \times 10^4$ HeLa cells in 96-wells, they were treated under each condition, cultured for 48 h, and washed with PBS. Fura-2 acetoxymethyl (2.5 μM: Dojindo Laboratories, Kumamoto, Japan) in DMEM medium was added to each well of a 96-well plate, and the plate was incubated at 37 °C for 30 min, and then washed twice with Tyrode solution containing (in mM) 140 NaCl, 5 KCl, 2 $CaCl_2$, 1 $MgCl_2$, 10 HEPES, and 10 D-glucose (pH adjusted to 7.4 with NaOH, and osmolality adjusted to 300 mosmol/kg $H_2O$ with D-mannitol). Plates were cultured at 37 °C for 1 h to stabilize cells in Tyrode solution. By exciting fura-2 fluorescence at 340/380 nm, intracellular $Ca^{2+}$ fluorescence signals were measured using a 510-nm filter. Fluorescence intensity was calculated as the F340/F380 ratio and shown as changes in the intracellular free $Ca^{2+}$ concentration.

**Co-immunoprecipitation and western blot analysis**. gTRPM7-deficient DT40 cells and those stably expressing hTRPM7-WT, -K1648R, or -Δ-kinase were grown on 10-cm dishes. Expression of hTRPM7 and its mutants was induced by adding 1 μg/ml tetracycline to the medium and culturing for 24–48 h. HEK293T cells were cultured for 48 h under each condition. DT40 or HEK293T cells were lysed in radioimmunoprecipitation assay buffer (pH 8.0) containing 0.1% SDS, 0.5% sodium deoxycholate, 1% Nonidet P40, 150 mM NaCl, 50 mM Tris-HCl, 1 mM PMSF, and 10 μg/ml leupeptin, and then centrifuged at 17,400 g for 15 min. Whole-cell lysates were fractionated by 7.5% SDS-PAGE and electrophoresed on a poly-vinylidene fluoride (PVDF) membrane. The blots were incubated with primary antibody anti-HA antibody (1:1000 dilution: 06340; Nacalai Tesque, Kyoto,

Japan) or anti-α-tubulin (1:2000 dilution: T6074; Sigma-Aldrich) as internal standard, and then incubated with secondary antibody of mouse IgG (1:3000 dilution, Amersham enhanced chemiluminescence (ECL): Amersham, Little Chalfont, UK) and HRP-linked whole sheep Ab (1:3000 dilution: NA931; Merck, Darmstadt, Germany). Co-immunoprecipitation was performed by using the Capturem™ IP & Co-IP Kit (Takara-Bio) and according to the manufacturer's instructions. Briefly, HEK293T cells were solubilized in Lysis/Equilibration Buffer containing Protease Inhibitor Cocktail, then centrifuged at 17,400 g for 20 min to remove cell debris. For co-immunoprecipitation, the cell lysate was incubated with anti-GFP polyclonal antibody (1:1000 dilution: 50430-2-AP; Proteintech) for 1 h. Then immunoprecipitated proteins were fractionated by 7.5% SDS-PAGE and electrotransferred onto a PVDF membrane. The blots were incubated with anti-mCherry monoclonal antibody (1:1000 dilution: GTX630189; GeneTex, Inc.) or anti-α-tubulin and visualized using the ECL system (Thermo Fisher Scientific). The chemiluminescence of the membrane was detected by the LAS system (LAS-3000; Fujifilm, Tokyo, Japan) and analyzed by the Image Gauge software (Fujifilm).

**Statistics and reproducibility**. All data are expressed as means ± SEM. We accumulated the data for each condition from at least three independent experiments. We evaluated statistical significance with Student's $t$ test for comparisons between two mean values. A value of $P < 0.05$ was considered significant. For Figs. 2c, e, 7b, e, 8e, g and Supplementary Fig. 2a, statistical differences of the data were evaluated with one-way analysis of variance followed by a Tukey's test for multiple comparisons and considered to be significant at $P < 0.05$. For Figs. 8b and 9b, the $\chi^2$ test was made to evaluate the association between two categorical variables with a value of $P < 0.05$ considered to be significant.

**Reporting summary**. Further information on research design is available in the Nature Research Reporting Summary linked to this article.

## Data availability

All the source raw data used for generating main Figures are provided as Supplementary data files and are available from the corresponding author on reasonable request.

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

## Acknowledgements

We thank Dr C. Schmitz and Dr A.-L. Perraud for providing DT40 cells and those expressing TRPM7 mutants and for advice on the culture technique of these cells. This work was supported in part by Grants-in-Aid for Scientific Research (KAKENHI) from the Japan Society for the Promotion of Science and the Ministry of Education, Culture, Sports, Science (Nos. 15K08197, 18K06864, JP20H05842 (Grant-in-Aid for Transformative Research Areas, "Dynamic Exciton")), and Central Research Institute of Fukuoka University (No. 177009).

## Author contributions

T.N. conducted all experiments and analysis. K.S.-N. conducted the PCR experiments. M.C.H. and Y.M. aided in the culture design, discussed on the data, and commented on the draft. T.N and Y.O. conceived and designed the work, wrote the manuscript, and approved the final version of manuscript.

## Competing interests

The authors declare no competing interests.
