## [Transparent Peer Review File · Communications Biology]

Reviewers' comments:

Reviewer #1 (Remarks to the Author):

In this manuscript, T. Numata et al. explored the functional connection between volume regulated anion channel (VRAC, also known as volume-sensitive outwardly rectifying chloride channel, or VSOR) and nonselective cation channel TRPM7. The broader physiological question behind this "technical" description is "How do cells sense and transduce cell volume changes to volume regulatory mechanisms?" In the case of VRAC/VSOR the field is still searching for answer(s).

The Authors conclude that TRPM7 protein is an ESSENTIAL regulator expression and activity of the LRRC8A-containing volume regulated anion channel (VRAC/VSOR) in two model cell lines (HeLa and chicken DT40). This conclusion is well justified by the provided experimental evidence, which is derived from expertly executed studies. The relevant findings represent a conceptual step forward and are of interest to the field.

The Authors further imply that the TRPM7 interacts at the molecular level (physically?) with VRAC/VSOR and that TRPM7's alpha-kinase domain is required for such an interaction, possibly via its scaffolding function. I have some questions related to the latter interpretation (see specific comments below).

Major-to-moderate issues:

[1] The Authors' assumption that alpha-kinase domain is required for VRAC/VSOR regulation comes with a caveat. Since the majority of delta-kinase-TRPM7 cells do not produce TRPM7 currents, it is possible that the TRPM7 channel activity (Ca²⁺ signal?) rather than alpha-kinase domain is required. Furthermore, the Authors state that [quote, lns 216-217] "In delta-kinase-expressing cells, VSOR currents similar in magnitude to those in WT cells were recorded in two of seven cells tested". For these reasons, I strongly suggest tempering conclusions on the critical necessity of the TRPM7 kinase domain.

[2] There is an inference in the manuscript that LRRC8A and TRPM7 physically interact. The Authors provide no evidence for this (such as co-IP, FRET, etc.) and need to stress that the above inference is a pure speculation at this time, which is yet to be experimentally tested.

Minor concerns:

[3] Methods: Please state upfront in the description of cell lines if the delta-kinase TRPM7 construct was overexpressed in TRPV7 KO cells ("rescue" expression) of WT cells. Depending on which variant of expression the Authors used, their interpretations of the data would be very different. My understanding is that overexpression has been performed in a "rescue" variation.

[4] The following statement (lns 212-215) is not fully supported by the presented data: "In WT hTRPM7-expressing cells, the amplitude of whole-cell VSOR currents was much larger than that of VSOR currents recorded in control DT40 cells". Side-by-side comparison in Fig. 7c does not show statistically significant difference. Please double-check your analyses (including correlative analysis in Fig. 5b, c) to establish if you have statistical grounds for your conclusion.

Reviewer #2 (Remarks to the Author):

In this work, the authors from Prof Okada's lab evaluate the role of swelling-induced activation of TRPM7 cation channels in regulatory volume decrease (RVD) mechanism of epithelial cells.

As it was demonstrated for others TRPs, TRPM7 might functionally interact with VRAC.

The work is interesting but requires major revision before it can be submitted.

In particular, the authors fail to demonstrate that the ability of TRPM-7 to modulate VRAC/LRRC8A expression and function is specific. The author should prove that the modulation of VRAC is not a side effect of having a functional, swelling induces, calcium entry.

Nonetheless the literature on VRAC and TRPs interaction and RVD is overlooked.

Specific comments

1) The observations that TRPM7 gene knockout abolish RVD in DT40 cell is in line with previous report from the same group. Overexpression could be due to an effect related to the increased path for swelling activated calcium entry. As claimed by the authors, many TRPs and TRPM belonging family member are sensitive to swelling/osmotic/volume stress. The authors have to show the same rescue/overexpression experiment showing the results they will obtain using at least one on the other TRP or better TRPM overexpression.

2) Electrophysiology:

a) all the values of liquid junction potential for all the ionic condition and solution tested must be included. In the analyses, how this value was calculated and taken into account must be included. The statistical analyses on E Rev calculation should be indicated.

3) Figure 3: the solution used must be indicated in the figure legend. The shift in the reversal potential should be evident by using different intracellular saline allowing to distinguish VSOR chloride current from TRPM polycationic current (See Nilius et al., 20...; Benfenati et al., 2007)

4) Figure 4 The bath solution used to test TRPM7 (line 342, 343 page 16) has a different (lower) ionic strength with respect to the one used to measure VSOR (line 333, 334 page 16). The authors must use a solutions with no magnesium, but with the same ionic strength, that is known to modulated VRAC conductance.

About this topic the following reference must be included (LRRC8 Proteins Form Volume-Regulated Anion Channels that Sense IonicStrength.

Syeda R, Qiu Z, Dubin AE, Murthy SE, Florendo MN, Mason DE, Mathur J, Cahalan SM, Peters EC, Montal M, Patapoutian A. *Cell*. 2016 Jan 28;164(3):499-511. doi: 10.1016/j.cell.2015.12.031.

Sabirov RZ, Prenen J, Tomita T, Droogmans G, Nilius B. *Pflugers Arch*. 2000 Jan;439(3):315-20. doi: 10.1007/s004249900186.

Reduction of ionic strength activates single volume-regulated anion channels (VRAC) in endothelial cells.

5) Figure 5a: lower pannel in which ionic condition was the E-Rev calculated? This is essential for understanding the time course behavior

6)Figure 7. Molecular Data: a western blot analyses to verify the level of protein expression of interest must be included.

7)Literature overlooking:

The authors are overlooking the literature. Among the other, the above mentioned must be included and also, at least:

B Nilius 1, J Prenen, U Wissenbach, M Bödding, G Droogmans

Differential Activation of the Volume-Sensitive Cation Channel TRP12 (OTRPC4) and Volume-Regulated Anion Currents in HEK-293 Cells. *Pflugers Arch* 2001 Nov;443(2):227-33.

doi: 10.1007/s004240100676.

Molecular composition and heterogeneity of the LRRC8-containing swelling-activated osmolyte channels in primary rat astrocytes.

Schober AL, Wilson CS, Mongin AA. *J Physiol.* 2017 Nov 15;595(22):6939-6951. doi: 10.1113/JP275053. Epub 2017 Sep 12.

LRRC8A is essential for swelling-activated chloride current and for regulatory volume decrease in astrocytes.

Formaggio F, Saracino E, Mola MG, Rao SB, Amiry-Moghaddam M, Muccini M, Zamboni R, Nicchia GP, Caprini M, Benfenati V.

FASEB J. 2019 Jan;33(1):101-113. doi: 10.1096/fj.201701397RR. Epub 2018 Jun 29. PMID: 29957062

Reviewer #3 (Remarks to the Author):

Comments to the authors

In this study, the authors have shown that TRPM7 is correlated with LRRC8A expression and regulates VSOR activity and RVD. This was obtained by the experiment with TRPM7 knockdown and knockout. It is really interesting that TRPM7 regulates VSOR via LRRC8A, however, the mechanism is not clear by this manuscript.

Major comments,

1. How is the expression of LRRC8A mRNA regulated by TRPM7? Directly or indirectly? The authors should discuss this point.

2. The authors concluded from the experiment with the mutant which deleted kinase domain that TRPM7 serves as an essential regulator of VSOR and LRRC8A expression, via its kinase domain. How does the kinase domain act on them? Please discuss it.

3. The authors mentioned about TRPM7 as an efficient target for clinical treatments in the end of discussion. How does TRPM7 inhibitor (NS8593 or 2-APB etc) act on VSOR activity and LRRC8A expression? They should try at least one compound.

4. In Fig 6b, the PCR products from different genes cannot be compared directly by optical density. gTRPM7 and hTRPM7 is homolog, not same. Because different primers must have different amplification efficiency at PCR reaction. Therefore, Fig6b should be removed.

Minor comments,

1. In Fig 6a, beta-actin is protein name. It should be replace to gene name (ACTB?).

2. In page 27, line 517, GAPDH should be ACTB.

Point-by-point responses to Reviewers' comments

Reviewers' comments:

Reviewer #1 (Remarks to the Author):

In this manuscript, T. Numata et al. explored the functional connection between volume regulated anion channel (VRAC, also known as volume-sensitive outwardly rectifying chloride channel, or VSOR) and nonselective cation channel TRPM7. The broader physiological question behind this “technical” description is “How do cells sense and transduce cell volume changes to volume regulatory mechanisms?” In the case of VRAC/VSOR the field is still searching for answer(s).

The Authors conclude that TRPM7 protein is an ESSENTIAL regulator expression and activity of the LRRC8A-containing volume regulated anion channel (VRAC/VSOR) in two model cell lines (HeLa and chicken DT40). This conclusion is well justified by the provided experimental evidence, which is derived from expertly executed studies. The relevant findings represent a conceptual step forward and are of interest to the field.

The Authors further imply that the TRPM7 interacts at the molecular level (physically?) with VRAC/VSOR and that TRPM7's alpha-kinase domain is required for such an interaction, possibly via its scaffolding function. I have some questions related to the latter interpretation (see specific comments below).

: We would like to thank the reviewer for relevant and helpful comments. According to the comments by the reviewer, we have revised the manuscript by carrying out additional experiments with presenting and discussing the data succinctly.

Major-to-moderate issues:

[1] The Authors' assumption that alpha-kinase domain is required for VRAC/VSOR regulation comes with a caveat. Since the majority of delta-kinase-TRPM7 cells do not produce TRPM7 currents, it is possible that the TRPM7 channel activity (Ca²⁺ signal?) rather than alpha-kinase domain is required.

Furthermore, the Authors state that [quote, Ins 216-217] “In delta-kinase-expressing cells, VSOR currents similar in magnitude to those in WT cells were recorded in two of seven cells tested”. For these reasons, I strongly suggest tempering conclusions on the critical necessity of the TRPM7 kinase domain.

: We have carried out following experiments to obtain more information on the role of TRPM7 kinase domain. First, we increased the number of experiments for VSOR currents in TRPM7- Δ -kinase-expressing gTRPM7 KO DT40 cells from 7 to 17. Only around 30% of these Δ -kinase-expressing cells were found to exhibit not only VSOR currents but also TRPM7 currents. Thus, the averaged VSOR and TRPM7 currents were both very much smaller than those in WT-TRPM7-expressing gTRPM7 KO DT40 cells. These results have been included in Fig. 8 (d-g) and the Results section (Lines 251-256; 264-269) and noted in the Discussion section (Lines 328-330) of the revised manuscript. Second, immunocytochemical studies showed that only 25% of TRPM7- Δ -kinase was found to be colocalized with LRRC8A at the plasma membrane in roughly consistent with the above functional TRPM7 and VSOR activity. This result has been shown in Fig. 9 (a,b) and noted in the Results section (Lines 272-279) and the Discussion section (Lines 328-331). Third, we examined the physical interaction between TRPM7 and LRRC8A by immunoprecipitation to investigate the involvement of VSOR regulation through the α -kinase domain of TRPM7. These results showed that TRPM7 and LRRC8A interact with each other through the α -kinase domain in a manner independent of the enzymatic activity of the α -kinase domain. These points have been included in Fig. 9c and the Results section (Lines 282-290) as well as in the Discussion section (Lines 330-339) of the revised manuscript.

[2] There is an inference in the manuscript that LRRC8A and TRPM7 physically interact. The Authors provide no evidence for this (such as co-IP, FRET, etc.) and need to stress that the above inference is a pure speculation at this time, which is yet to be experimentally tested.

: As the referee pointed out, there was no evidence for the interaction between LRRC8A and TRPM7 in the previous manuscript. In the present manuscript, we have provided the evidence obtained by immunochemical studies

and by co-immunoprecipitation. These experiments showed that LRRC8A becomes co-localized at the plasma membrane and is co-precipitated with WT TRPM7 and its K1648R mutant, but little with its Δ -kinase mutant. These results indicate the involvement of TRPM7 in VSOR regulation through interaction with the α -kinase domain of TRPM7 in a manner independent of enzymatic activity of the α -kinase domain. These data have been presented in Fig. 9 and described in the Results section (Lines 272-290) as well as in the Discussion section (Lines 330-339; 366-371) in the revised manuscript.

Minor concerns:

[3] Methods: Please state upfront in the description of cell lines if the delta-kinase TRPM7 construct was overexpressed in TRPV7 KO cells (“rescue” expression) of WT cells. Depending on which variant of expression the Authors used, their interpretations of the data would be very different. My understanding is that overexpression has been performed in a “rescue” variation.

: As the referee pointed out, in the experiments for Figs. 6c, 7 and 8 (previous Figs. 5c, 6 and 7), we made “rescue” or “complementary” expression of hTRPM7 or its mutants in gTRPM7-KO DT40 cells. We have thus revised with making exact descriptions such as “gTRPM7-deficient DT40 cells complemented with hTRPM7” throughout the revised manuscript (Lines 196-217; 223-271; 317-321; 326-330; 354-357; 397-402; Legends for Figs. 6c, 7 and 8).

[4] The following statement (lns 212-215) is not fully supported by the presented data: “In WT hTRPM7-expressing cells, the amplitude of whole-cell VSOR currents was much larger than that of VSOR currents recorded in control DT40 cells”. Side-by-side comparison in Fig. 7c does not show statistically significant difference. Please double-check your analyses (including correlative analysis in Fig. 5b, c) to establish if you have statistical grounds for your conclusion.

: Statistical analyses were re-performed between the data shown in Fig. 8e (previous Fig. 7c) and Fig. 3b. We found a significant difference between the mean amplitude of whole-cell VSOR currents observed at +100 mV in control

DT40 cells (presented in Fig. 3b) and in WT hTRPM7-expressing gTRPM7-deficient DT40 cells (presented in Fig. 8e). This fact has been noted in the Results section (Lines 248-251) in the revised manuscript. We also checked a statistical difference between the slopes shown in Figs. 6b and 6c (previous Figs. 5b and 5c) by the χ^2 test and found no significance. Thus, we have corrected the related sentence (Lines 200-204) by removing a word “steeper” in the revised manuscript.

Reviewer #2 (Remarks to the Author):

In this work, the authors from Prof Okada's lab evaluate the role of swelling-induced activation of TRPM7 cation channels in regulatory volume decrease (RVD) mechanism of epithelial cells.

As it was demonstrated for others TRPs, TRPM7 might functionally interact with VRAC.

The work is interesting but requires major revision before it can be submitted.

In particular, the authors fail to demonstrate that the ability of TRPM-7 to modulate VRAC/LRRC8A expression and function is specific. The author should prove that the modulation of VRAC is not a side effect of having a functional, swelling induces, calcium entry.

Nonetheless the literature on VRAC and TRPs interaction and RVD is overlooked.

: We would like to thank the reviewer for relevant and helpful comments. According to the comments by the reviewer, we have revised the manuscript by carrying out additional experiments and discussing the data succinctly.

Specific comments

1) The observations that TRPM7 gene knockout abolish RVD in DT40 cell is in line with previous report from the same group. Overexpression could be due to an effect related to the increased path for swelling activated calcium entry. As claimed by the authors, many TRPs and TRPM belonging family member are sensitive to swelling/osmotic/volume stress. The authors have to show the same

rescue/overexpression experiment showing the results they will obtain using at least one on the other TRP or better TRPM overexpression.

: Since TRPM4 is known to be another mechanosensitive TRPM member and was found to be endogenously expressed in HeLa cells (Supplementary Fig 1c), we investigated its regulatory role in VSOR activity. To our surprise, as shown in Supplementary Fig. 1, siRNA-mediated knockdown of TRPM4 was found to affect neither the VSOR currents (a, b) nor the endogenous expression of hLRRC8A (c) in HeLa cells. In addition, overexpression of hTRPM4 failed to rescue VSOR activity in gTRPM7-deficient DT40 cells, as shown in Supplementary Fig. 2a. Furthermore, co-immunoprecipitation studies excluded the possibility of a physical interaction between TRPM4 and LRRC8A, as shown in Supplementary Fig. 2b. These results of TRPM4 are in contrast to those of TRPM7 and have been described in the Results section (Lines 112-116; 246-247; 283-285) of the revised manuscript

2) Electrophysiology:

a) all the values of liquid junction potential for all the ionic condition and solution tested must be included. In the analyses, how this value was calculated and taken into account must be included. The statistical analyses on E_{rev} calculation should be indicated.

: According to the referee comments, we have described how the liquid junction potential was calculated and taken into account and all the values of liquid junction potential were given in the Methods section (Lines 425-429; 442-443; 456-457; 466-467) of the revised manuscript. Also, we have added the mean peak (positive) and mean bottom (negative) values of the E_{rev} shift observed in the ramp-clamp experiments shown in Fig. 6a as well as the result of statistical analysis for these mean E_{rev} values in the Results section (Lines 173-180) of the revised manuscript.

3) Figure 3: the solution used must be indicated in the figure legend. The shift in the reversal potential should be evident by using different intracellular saline allowing to distinguish VSOR chloride current from TRPM polycationic current (See Nilius et al., 20...; Benfenati et al., 2007)

: In the experiments for Fig. 4 (previous Fig. 3), we eliminated ATP and most Cl⁻ ions from the intracellular solution to prevent possible VSOR Cl⁻ current contamination. This fact has been noted not only in the Methods section (Lines 451-453) but also in the Legend for Fig. 4 (Lines 656-658) of the revised manuscript.

4) Figure 4 The bath solution used to test TRPM7 (line 342, 343 page 16) has a different (lower) ionic strength with respect to the one used to measure VSOR (line 333, 334 page 16). The authors must use a solutions with no magnesium, but with the same ionic strength, that is known to modulated VRAC conductance.

: VSOR activity is affected by changes in ionic strength and magnesium concentration in the intracellular, but not extracellular (bath), solution. For VSOR and TRPM7 current measurements, we used intracellular (pipette) solutions with the ionic strength of ~130 and ~110 mM, respectively. These values have been included in the Methods section (Lines 442 & 450). Since VSOR activation is known to be induced or facilitated by reduction of intracellular ionic strength, we eliminated ATP and most Cl⁻ ions from the intracellular solution employed for TRPM7 current measurements to prevent possible VSOR Cl⁻ current contamination. Such has been noted in the Methods section (Lines 451-453) and in Legend for Fig. 4 (Lines 657-659).

About this topic the following reference must be included (LRRC8 Proteins Form Volume-Regulated Anion Channels that Sense IonicStrength.

Syeda R, Qiu Z, Dubin AE, Murthy SE, Florendo MN, Mason DE, Mathur J, Cahalan SM, Peters EC, Montal M, Patapoutian A. Cell. 2016 Jan 28;164(3):499-511. doi: 10.1016/j.cell.2015.12.031.

Sabirov RZ, Prenen J, Tomita T, Droogmans G, Nilius B. Pflugers Arch. 2000 Jan;439(3):315-20. doi: 10.1007/s004249900186.

Reduction of ionic strength activates single volume-regulated anion channels (VRAC) in endothelial cells.

: We have revised the Methods section (Lines 451-453) of our manuscript with citing these papers together with several other relevant papers.

5) Figure 5a: lower panel in which ionic condition was the E-Rev calculated?
This is essential for understanding the time course behavior

: According to the referee comment, we have described detailed ionic conditions employed in the experiments for Fig. 6 (previous Fig. 5) in the Methods section (Lines 459-467) of the revised manuscript. In this experiments, cationic TRPM7 currents and anionic VSOR currents were simultaneously measured at the equilibrium potentials for anions and cations, -30 and +30 mV, respectively. Such has been clarified in the Results section (Lines 171-173) and Legend for Fig. 6 (Lines 686-695) of the revised manuscript.

6) Figure 7. Molecular Data: a western blot analyses to verify the level of protein expression of interest must be included.

: According to the referee comment, we made the additional western blot experiments, as described in the Methods section (Lines 553-577) in the revised manuscript. The data thus obtained have been included in Fig. 8 (previous Fig. 7) as Fig. 8c in the revised manuscript. These data have been described in the Results section (Lines 236-242).

7)Literature overlooking:

The authors are overlooking the literature. Among the other, the above mentioned must be included and also, at least:

B Nilius 1, J Prenen, U Wissenbach, M Bödding, G Droogmans
Differential Activation of the Volume-Sensitive Cation Channel TRP12 (OTRPC4) and Volume-Regulated Anion Currents in HEK-293 Cells. Pflugers Arch 2001 Nov;443(2):227-33.
doi: 10.1007/s004240100676.

Molecular composition and heterogeneity of the LRRC8-containing swelling-activated osmolyte channels in primary rat astrocytes.
Schober AL, Wilson CS, Mongin AA. J Physiol. 2017 Nov 15;595(22):6939-6951.
doi: 10.1113/JP275053. Epub 2017 Sep 12.

LRRC8A is essential for swelling-activated chloride current and for regulatory volume decrease in astrocytes.

Formaggio F, Saracino E, Mola MG, Rao SB, Amiry-Moghaddam M, Muccini M, Zamboni R, Nicchia GP, Caprini M, Benfenati V.

FASEB J. 2019 Jan;33(1):101-113. doi: 10.1096/fj.201701397RR. Epub 2018 Jun 29.

PMID: 29957062

: According to the referee comment, we have included these literatures (as Refs. 59, 27 and 26) in the revised manuscript.

Reviewer #3 (Remarks to the Author):

Comments to the authors

In this study, the authors have shown that TRPM7 is correlated with LRRC8A expression and regulates VSOR activity and RVD. This was obtained by the experiment with TRPM7 knockdown and knockout. It is really interesting that TRPM7 regulates VSOR via LRRC8A, however, the mechanism is not clear by this manuscript.

: We would like to thank the reviewer for relevant and helpful comments. According to the comments by the reviewer, we have revised the manuscript by carrying out additional experiments and discussing the data succinctly.

Major comments,

1. How is the expression of LRRC8A mRNA regulated by TRPM7? Directly or indirectly? The authors should discuss this point.

: To respond to this referee comment, we conducted additional experiments on the involvement of Ca²⁺ influx via TRPM7 in the regulation of LRRC8A mRNA expression by observing the effects of two-day exposure to a TRPM7 blocker, NS8593, or a Ca²⁺ chelator, EGTA, in HeLa cells. Such long-time extracellular

application of NS8593 or EGTA was found to reduce not only TRPM7 currents and the intracellular free Ca²⁺ concentration but also the expression of LRRC8A mRNA and VSOR currents. Thus, it appears that TRPM7-induced regulation of LRRC8A mRNA expression is indirectly attained by steady-state Ca²⁺ influx via TRPM7 channels. These results have been presented in Fig. 2 and described in the Results section (Lines 117-130) and mentioned in the Discussion section (Lines 363-371).

2. The authors concluded from the experiment with the mutant which deleted kinase domain that TRPM7 serves as an essential regulator of VSOR and LRRC8A expression, via its kinase domain. How does the kinase domain act on them? Please discuss it.

: As noted above, regulation of the LRRC8A mRNA expression by TRPM7 is mediated by steady-state Ca²⁺ influx through TRPM7 channels.

To respond to this referee comment, we further made additional immunochemical and co-immunoprecipitation experiments to examine roles of the α -kinase domain of TRPM7 in the plasmalemmal expression of LRRC8A protein and in the physical interaction between TRPM7 and LRRC8A, respectively.

First, in HeLa cells exposed to a hypotonic solution, transfected LRRC8A-mCherry was found to become localized to the periphery (plasmalemmal-like) region together with WT TRPM7-EGFP or the kinase activity-lacking mutant K1648R-EGFP but not much with the EGFP-conjugated Δ -kinase mutant. These results suggest that TRPM7 facilitates the plasmalemmal expression of LRRC8A protein via the kinase domain of TRPM7, in a manner independent of the enzyme activity. These data have been presented in Fig. 9 (a, b) and described in the Results section (Lines 272-279) as well as in the Discussion section (Lines 327-337).

Next, in osmotically swollen HEK293T cells co-transfected with LRRC8A and WT TRPM7, the K1648 mutant or the Δ -kinase mutant, LRRC8A was found to be co-precipitated with WT TRPM7 or the K1648R mutant but not much with the Δ -kinase mutant. These results strongly suggest that LRRC8A physically interacts with TRPM7 via the C-terminal kinase domain of TRPM7, in a manner independent of its enzyme activity, after osmotic cell swelling. These data have

been presented in Fig. 9c and described in the Results section (Lines 279-290) as well as in the Discussion section (Lines 331-335).

3. The authors mentioned about TRPM7 as an efficient target for clinical treatments in the end of discussion. How does TRPM7 inhibitor (NS8593 or 2-APB etc) act on VSOR activity and LRRCA8A expression? They should try at least one compound.

: According to this referee comment, we made additional experiments using a TRPM7 inhibitor, NS8593. In HeLa cells, exposure to NS8593 for two days was found to reduce not only TRPM7 currents and the intracellular free Ca^{2+} concentration but also LRRCA8A mRNA expression and VSOR currents to the extent similar to the suppressing effects of TRPM7 knockdown. These results have been presented in Fig. 2 and described in the Results section (Lines 117-130) as well as in the Discussion section (Lines 315-317) in the revised manuscript.

4. In Fig 6b, the PCR products from different genes cannot be compared directly by optical density. gTRPM7 and hTRPM7 is homolog, not same. Because different primers must have different amplification efficiency at PCR reaction. Therefore, Fig6b should be removed.

: We agree with this referee comment. The previous Fig. 6b has been deleted from the corresponding figure (Fig. 7) in the revised manuscript.

Minor comments,

1. In Fig 6a, beta-actin is protein name. It should be replace to gene name (ACTB?).

: According to this referee's indication, we have replaced "beta-actin" with the gene name "ACTB" throughout the revised manuscript (Lines 487, 494, 704 & 706).

2. In page 27, line 517, GAPDH should be ACTB.

: According to this referee comment, we have revised the description by replacing “GAPDH” with “ACTB” (Lines 712)

REVIEWERS' COMMENTS:

Reviewer #1 (Remarks to the Author):

This reviewer is satisfied with the Authors' responses. The additional data included with the revised manuscript fortify the Authors' conclusions and address previous concerns.

Reviewer #2 (Remarks to the Author):

The work is improved in its quality and content. The review was punctual and accurate. The work can be published without further revision.

Reviewer #3 (Remarks to the Author):

The authors have answered all of this reviewer's comments accurately. The present manuscript is now appropriate for the publication to this journal.